biomedical engineering/immunology/ systems theory

virus therapy, control theory and applications, oncolytic virus, viral injection, feedback control, personalized tumour therapy

**Author for correspondence:**
Pablo S. Rivadeneira
e-mail: psrivade@unal.edu.co

# Oncolytic virus therapy benefits from control theory

Anet J. N. Anelone[1], María F. Villa-Tamayo[2] and Pablo S. Rivadeneira[2]

[1]School of Mathematics and Statistics, The University of Sydney, Camperdown, New South Wales 2006, Australia
[2]Universidad Nacional de Colombia, Facultad de Minas, Grupo GITA, Cra. 80 No 65-223, Medellín, Colombia

 AJNA, 0000-0002-7481-8134; MFV-T, 0000-0002-0839-4070; PSR, 0000-0001-8392-4556

Oncolytic virus therapy aims to eradicate tumours using viruses which only infect and destroy targeted tumour cells. It is urgent to improve understanding and outcomes of this promising cancer treatment because oncolytic virus therapy could provide sensible solutions for many patients with cancer. Recently, mathematical modelling of oncolytic virus therapy was used to study different treatment protocols for treating breast cancer cells with genetically engineered adenoviruses. Indeed, it is currently challenging to elucidate the number, the schedule, and the dosage of viral injections to achieve tumour regression at a desired level and within a desired time frame. Here, we apply control theory to this model to advance the analysis of oncolytic virus therapy. The control analysis of the model suggests that at least three viral injections are required to control and reduce the tumour from any initial size to a therapeutic target. In addition, we present an impulsive control strategy with an integral action and a state feedback control which achieves tumour regression for different schedule of injections. When oncolytic virus therapy is evaluated *in silico* using this feedback control of the tumour, the controller automatically tunes the dose of viral injections to improve tumour regression and to provide some robustness to uncertainty in biological rates. Feedback control shows the potential to deliver efficient and personalized dose of viral injections to achieve tumour regression better than the ones obtained by former protocols. The control strategy has been evaluated *in silico* with parameters that represent five nude mice from a previous experimental work. Together, our findings suggest theoretical and practical benefits by applying control theory to oncolytic virus therapy.

# Author summary

Oncolytic virus therapy benefits from control theory for the following reasons:

1. Controllability and accessibility tests demonstrate that viral injections are appropriate impulses to control tumour regression, and for this specific model, at least three injections are required to control and reduce the tumour from any initial size to a therapeutic target.
2. Control theory proposes an analytical framework to achieve tumour regression using a feedback control, which when evaluated *in silico* automatically tunes the dose of viral injections.
3. Control theory is able to personalize dosages for the viral injections of a given subject to meet his therapeutic objectives and constraints.
4. Feedback control provides insights on tumour–virus interactions during therapy.
5. Feedback control has the potential to deliver therapies which exhibit some robustness to uncertainty and perturbation in the kinetic parameters of the tumour and the virus.
6. Control theory shows the potential to deliver efficient total doses for therapies without feedback control to achieve tumour regression using personalized or common dosages of viral injections.

# 1. Introduction

Viruses preferentially target particular types of cells and viral replication often destroys infected cells via lysis and this produces new viral particles *in vivo* [1–4]. Oncolytic viruses are such viruses which preferentially infect and lyse tumour cells due to extensive viral replication inside these cells [2,4,5]. Oncolytic virus therapy refers to clinical applications of oncolytic viruses to eradicate or at least reduce tumours [5].

Oncolytic virus therapy is a promising cancer therapy for various reasons. One of its main advantages is that oncolytic viruses can be engineered to target specific tumour cells without damaging healthy cells [2,4–6]. Different oncolytic viruses have been genetically engineered to take advantage of different virus pathogenesis and properties [5,7]. For example, oncolytic virus therapies have been conducted using adenoviruses and measles viruses [5,7–10]. Thus, oncolytic virus therapy stands as a promising solution which can be used alone or in combination with surgery or other anti-tumour treatments to reduce or eradicate tumour [6,7]. Oncolytic virus therapy has been successful for some cancer patients; however, it is not clear how to design protocols to achieve appropriate outcomes in all cases [5,7,11].

Important advances on oncolytic virus therapy were made using mathematical modelling of within-host dynamics of oncolytic viruses in animal studies [10–14]. Mathematical modelling of oncolytic virus experiments has delivered qualitative and quantitative insights on the interactions between the tumour, cell cycles, the oncolytic virus and immune responses *in vivo* [5,11,12,14]. These models are useful to understand and optimize the impact of different factors such as changes in the genetic of the virus, in the dosage or in the scheduling of injections. The mathematical model in [14] has been used to show that adenoviruses can be optimized by genetic modifications to improve tumour reduction. The results in [5,14] have also shown variations in the time course of the total tumour volume and variations in biological rates despite the fact that the experiments were conducted in genetically identical mice. In addition, the experimental protocol for viral injections did not achieve similar tumour reduction even though the same virus and the same protocol were used for each mouse. Indeed, it is challenging to achieve desired therapeutic goals in the presence of uncertainties in biological rates and processes.

These challenges motivate the application of control theory to improve understanding and outcomes of oncolytic virus therapy. Control theory is a mature discipline in the study of uncertain and nonlinear dynamical processes [15–17]. Interdisciplinary studies between virus dynamics and control theory have advanced our understanding of virus dynamics, immune responses and treatments to enforce a healthy state [15,18–22]. The immune system can be analysed as a closed-loop system in which immune responses operate as intrinsic control inputs and drugs operate as extrinsic control inputs. Previous studies have found synergies between the dynamics of biomedical processes and control schemes. For instance, the T cell response is similar to switched control schemes such as sliding mode control theory [21–23]. In addition, the sliding mode *reachability condition* has been used to formulate dynamical conditions for the containment of HIV infection by the CD8+ T cell response and antiretroviral drugs [21–23]. Furthermore, antiretroviral treatments have been analysed as impulsive

control strategies in which drug uptakes are impulses to force HIV loads to reach and remain at undetectable levels [18,24,25]. An impulsive control approach has also been used in the context of type-1 diabetes to regulate the injection of insulin [26,27]. The results highlight that the impulsive control scheme delivers reasonable treatment regimes in both contexts to achieve therapeutic goals despite parameter and modelling uncertainties [24,25,27,28]. Thus, it is sensible to apply control theory for oncolytic virus therapy to investigate effective and robust protocols to sustain tumour regression.

In this paper, we build on previous work by performing a control analysis of the calibrated mathematical model of oncolytic virus therapy developed in [11,14]. This model describes the within-host dynamics of breast cancer cells infected by oncolytic adenoviruses; the model fits well the time course of the total tumour volume observed during experimental treatments on nude mice [5,14]. We contribute to current mathematical and experimental findings by implementing a novel method in which we cast the problem of tumour regression during therapy as a control challenge. In fact, it has been shown that changes in virus dynamics, i.e. the input, lead to changes in the total tumour volume, i.e. the output [5,14]. We analysed the within-host dynamics of oncolytic virus therapy as a closed-loop system in which viral injections at given days represent an impulsive control strategy to achieve tumour regression. We used controllability and accessibility tests (see [25,29,30]), to determine whether the tumour can be steered from any initial level to a desired level in a neighbourhood of a system equilibrium. These tests also determine the minimum number of injections required to control and reduce the tumour from any initial level to a desired therapeutic target. Although this minimum number is coherent with the experimental protocol given in [5], this result is valid for this particular model and deserves further experimental research. Furthermore, we developed a state feedback control with an integral action to personalize and tune the dose of viral injection and to ensure that tumour regression exhibits some robustness of uncertainty in biological rates. Feedback control of the tumour also delivers insights on tumour–virus interactions affecting the performance of therapies. The control strategy is *in silico* tested and compared with the experimental protocol used in [5] for five nude mice. Together, the findings present compelling benefits from applying control theory to oncolytic virus therapy.

## 2. Context

We conducted our analysis in the context of the experimental studies in [5] where genetically engineered oncolytic adenoviruses are used to reduce the total number of breast cancer cells in nude mice within 60 days. Since nude mice do not have an immune system, tumour regression is only due to the oncolytic virus therapy. The experiments started with about 90–300 tumour cells to test an oncolytic PEG-modified adenovirus conjugated with herceptin (Ad-PEG-HER). Ad-PEG-HER was shown to be the best at tumour regression in the experiments [5,14]. Each experiment followed a standard protocol of $10^{10}$ viral particles injected at days 0, 2 and 4.

The interactions between oncolytic virus and tumour cells are mathematically described according to the work in [14] by a set of three ordinary differential equations (ODEs) of the form $\dot{\xi} = f(\xi, u)$ as

and

$$
\left.\begin{aligned}
\frac{dS(t)}{dt} &= r \ln\left(\frac{L}{S(t)}\right) S(t) - \frac{\beta S(t) V(t)}{S(t) + I(t) + \varepsilon}, \\
\frac{dI(t)}{dt} &= \frac{\beta S(t) V(t)}{S(t) + I(t) + \varepsilon} - d_I I(t) \\
\frac{dV(t)}{dt} &= u(t) - d_V V(t) + \alpha d_I I(t),
\end{aligned}\right\}
\tag{2.1}
$$

where the state $\xi$ is composed by the number of susceptible tumour cells, $S$, the number of tumour infected cells, $I$ and the number of virus particles ($\log_{10}$), $V$, i.e. $\xi = (S\ I\ V)'$. The output of the model corresponds to the total number of tumour cells $T = S + I$, and it is measured every 2 days. The parameters of the model are obtained from the best-fit of the model (2.1) to experimental data from [5]; see appendix, Methods, and table 1.

The input $u$ is the control action because $u$ corresponds to the viral load $u_V$, injected at predefined days, $t_k$

$$
u(t) = \begin{cases} u_V, & t = \tau_k, \ k \in \mathbb{N} \\ 0, & \text{otherwise} \end{cases}.
\tag{2.2}
$$

**Table 1.** Initial conditions and parameter values of the model (2.1) for Ad-PEG-HER.

| symbol | units | S1 | S2 | S3 | S4 | S5 |
|---|---|---|---|---|---|---|
| $S_0$ | cells | 238.3535 | 200.0340 | 101.5400 | 140.3436 | 128.1481 |
| $I_0$ | cells | 0 | 0 | 0 | 0 | 0 |
| $V_0$ | virus | 0 | 0 | 0 | 0 | 0 |
| $r$ | $day^{-1}$ | 0.0378 | 0.0733 | 0.0224 | 0.0316 | 0.0603 |
| $L$ | $cells \times 10^6$ | 8466.8 | 3179.1 | 4922.4 | 8317.1 | 936.4293 |
| $\beta$ | $day^{-1}$ | 1.12 | 1.4987 | 0.2 | 1.2108 | 1.3606 |
| $d_I$ | $day^{-1}$ | 2 | 1.9995 | 2 | 0.1 | 0.1 |
| $d_v$ | $day^{-1}$ | 2.0872 | 3.2287 | 3.5 | 1.8730 | 1.8416 |
| $\alpha$ | $virus \times 10^9$ | 2 | 2.0015 | 2 | 3.7748 | 3.7541 |

**Table 2.** Results of controllability tests for the model (2.1) of oncolytic virus therapy.

| steady state | controllable? |
|---|---|
| $Eq_h = (0\ 0\ 0;\ 0)$ (A 1) | no |
| $Eq_e = (L\ 0\ 0;\ 0)$ (A 2) | yes |
| $Eq_v = (S_{ss}\ I_{ss}\ V_{ss};\ 0)$ (A 3) | yes |

We considered that the objective of the therapy is to reduce and maintain the total number of tumour cells below 50 cells within 60 days.

# 3. Results

## 3.1. The tumour is accessible and controllable from any initial size by oncolytic virus therapy

First of all, we addressed the fundamental question: 'Is the tumour accessible by oncolytic virus therapy?' The accessibility test is a theoretical tool to determine whether there exists a control sequence to steer a nonlinear system from any initial state to another state in finite time [25,29,30]. The mathematical analysis suggests that the strictly subspace $\mathbb{R}^3_+$ is accessible for any $S > 0$, and $S \neq Le^{(-1+d_i/r)}$; see Methods. Furthermore, we evaluated the impulse relative degree to determine that at least three injections are required to reach any final state in the strictly positive subspace of $\mathbb{R}^3$; see Methods. Therefore, the total number of tumour cells can indeed be modified by impulsive inputs corresponding to the viral loads.

As we aim to propose a linear control strategy, we checked that the linearized model is controllable too. Controllability is a structural property with the same interpretability for linear systems as the accessibility for the nonlinear case. We formally answered this question by testing the controllability of the model (2.1); see Methods. The analytical results indicate that the tumour is controllable by viral injections at its steady-state carrying capacity and at another non-zero steady state below the carrying capacity; see table 2. In addition, since the tumour is controllable, the controllability test establishes that at least three injections are required to reduce the tumour to a therapeutic equilibrium; see Methods. These results suggest that viral injections are suitable control inputs to reduce the tumour from any initial level to a desired level in finite time.

## 3.2. Synergies between oncolytic virus therapy and impulsive control theory

To our knowledge, there is currently no analytical framework to design efficient oncolytic virus therapy. Therefore, we investigated a control scheme corresponding to oncolytic virus therapy; see the appendix. Since the oncolytic virus is injected in the body at different days [5], the injection of virus is considered as

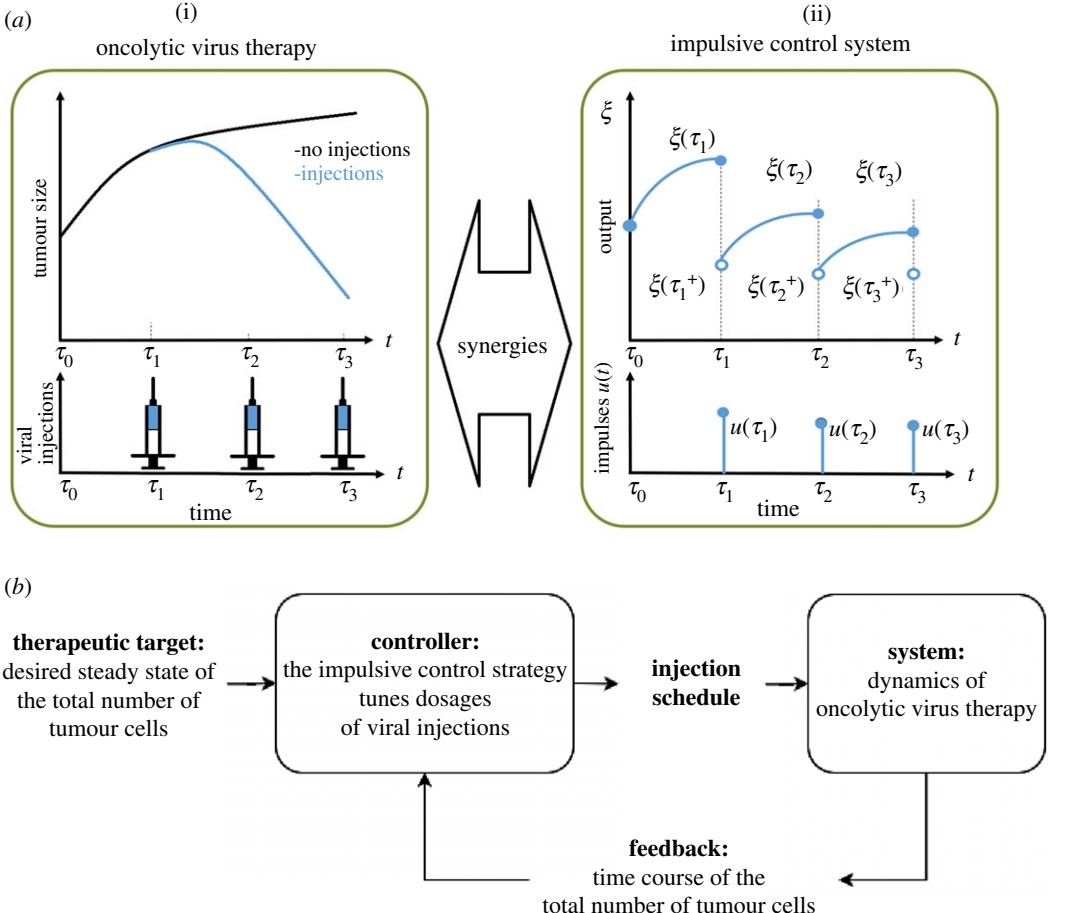

**Figure 1.** Synergies between oncolytic virus therapy and impulsive control theory. (a) (i) Example of oncolytic virus therapy. The tumour is controlled by injecting an oncolytic virus at different times. (a) (ii) Example of an impulsive control system. The output signal $h(\xi)$ is controlled by applying impulses, $u(t)$ at different times. (b) Block diagram of oncolytic virus therapy with impulsive control theory.

a discontinuous control action that has an amplitude with negligible duration and remains null the rest of the sampling time; see figure 1. Thus, the therapy is characterized by two responses: a forced response at the moment of viral injection, and a free response in the absence of viral injection. Similarly, impulsive control systems are those in which the input is of very short duration in relation to the sample time, so its action time is considered negligible. An impulsive control system is characterized by two responses: a forced response at the moment of input action, and a free response when the input is zero. The behaviour of such system is illustrated in figure 1, where $\xi(\tau_k)$ is the state before the impulse action, and $\xi(\tau_k^+)$ is the state after the control action $u$ at times $\tau_k$, $k \in \mathbb{N}$. This suggests synergies between impulsive control systems and oncolytic virus therapies with discrete-time viral injections. Thus, an impulsive control strategy could be designed to perform oncolytic virus therapy.

Consequently, we investigated an impulsive control strategy for oncolytic virus therapy; see the appendix. To that aim, we considered the model (2.1) as an impulsive control system where viral injections represent impulses applied at some time instants. We established a feedback control law using the total number of tumour cells, and we designed it in such a way that dosages of viral injection bring the total number of tumour cells to a predefined therapeutic target in a desired time; see figures 1 and 2. Together, these findings suggest that oncolytic virus therapy could benefit from analytical tools from impulsive control theory to achieve tumour regression.

## 3.3. Control theory delivers efficient and personalized viral injections

It is currently challenging to formulate efficient oncolytic virus therapy [5,6,14]. Therefore, we investigated personalized therapies by applying impulsive control theory; see the appendix. We used the schedule of one injection at days 0, 2 and 4 from the experimental therapy with Ad-PEG-HER in

**Figure 2.** Guidelines to perform oncolytic virus therapy using the proposed impulsive control strategy.

[5], and we compared the outcomes of the personalized and experimental therapies. The experimental therapy fails to decrease the tumour to the therapeutic zone, i.e. less than or equal to 50 total tumour cells in all subjects; see figure 3. The tumour keeps increasing in subjects 2 and 3 whereas the tumour decreases after an initial peak in subjects 1, 4 and 5. Thus, the experimental therapy performs differently among subjects, suggesting that delivering personalized viral injections could improve the outcomes of therapy. The personalized therapy reduces the tumour within the therapeutic zone faster than the experimental therapies; see figure 3. The tumour rebounds few days before the end of the follow-up in subjects 2, 3 and 5, suggesting that additional injections or a feedback control could be useful to keep the tumour within the therapeutic zone. Although the personalized therapies tend to decrease the dosage of viral injections over time, the personalized therapies apply higher dosages than the experimental therapy; see figure 3. Consequently, viral loads are higher in the personalized therapy than the experimental therapy during the period of injections from day 0 to 4; see figure 3. Nevertheless, viral loads tend to become lower in the personalized therapy than the experimental therapy after the last injection, suggesting a rapid clearance of viral loads which might be toxic; see figure 3. Together, these results suggest that applying control theory to oncolytic virus therapy is beneficial to deliver personalized dosages of viral injections to achieve therapeutic objectives.

## 3.4. Feedback control tunes viral injections to achieve and sustain tumour regression under different constraints

Next, we aimed to allow oncolytic virus therapy to benefit from feedback control to sustain tumour regression. We investigated the outcomes of personalized therapies with respect to changes in the frequency of injections. We used the feedback control strategy in figure 2 to personalize doses and to configure the controller so that the total number of tumour cells reach and remain inside the therapeutic zone from day 15 onward; see the appendix. And, we assessed the outcomes of this configuration when one injection is made every day, every 2 days, every 5 days and every 10 days. Changes in the schedule of viral injections tend to influence the time course of tumour, the viral load

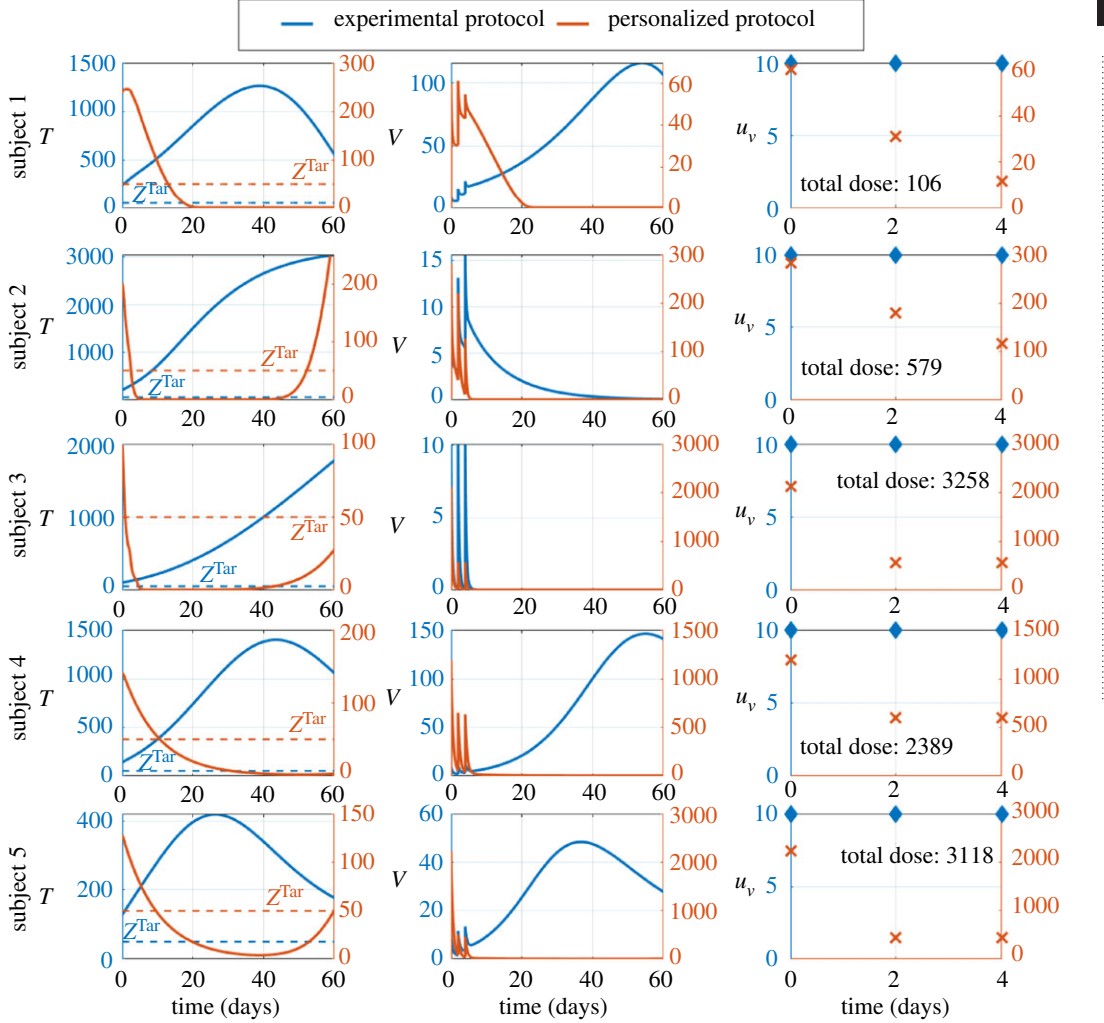

**Figure 3.** Control theory delivers efficient and personalized viral injections. Simulations of oncolytic virus therapies with Ad-PEG-HER. The left axes, blue lines and blue diamonds refer to the experimental therapy. The right axes, red lines and red crosses refer to the personalized therapy. Each row shows the outcomes of therapies in a given subject. The left column shows the time course of the total number of tumour cells. The dashed lines and $Z^{Tar}$ refer to the therapeutic zone, i.e. less than 50 for the total number of tumour cells. The middle column shows the time course of the viral loads *in vivo*. The right column shows the time course of viral injections. The 'total dose' refers to the total dose of the personalized therapy for the corresponding subject. The total dose for the experimental therapy is 30 in all subjects.

and the dosage during therapy; see figure 4. When the time between injections increases, the feedback control tends to increase the initial dose and subsequent doses at which all injection schedules match, suggesting that this increase in dosages aims to compensate for the longer time between injections; see figure 4. When the number of days between injections increases, the feedback control tends to improve tumour regression during the personalized therapies, supporting the notion that few high doses tends to perform better than multiple low doses [14]; see figure 4. Changes in the schedule of viral injections tend to have different impacts on the time course of the tumour among subjects, reflecting the heterogeneity in virus–tumour dynamics among subjects; see figure 4. When injections are made every day or every 2 days, tumour regression tends to exhibit an exponential decline with or without an initial peak; see figure 4. When injections are made every 5 or 10 days, the tumour tends to exhibit decreasing oscillations because the tumour rebounds between injections; see figure 4. Nevertheless, the total number of tumour cells reach and remain inside the therapeutic zone from day 15 onwards despite changes in the schedule of injections, suggesting that doses are tuned appropriately by the controller to overcome changes in the time between injections; see figure 4. When the time between injections increases, the total viral load administered tends to increase in subjects 3 and 5, supporting the notion of higher doses to compensate for longer time between

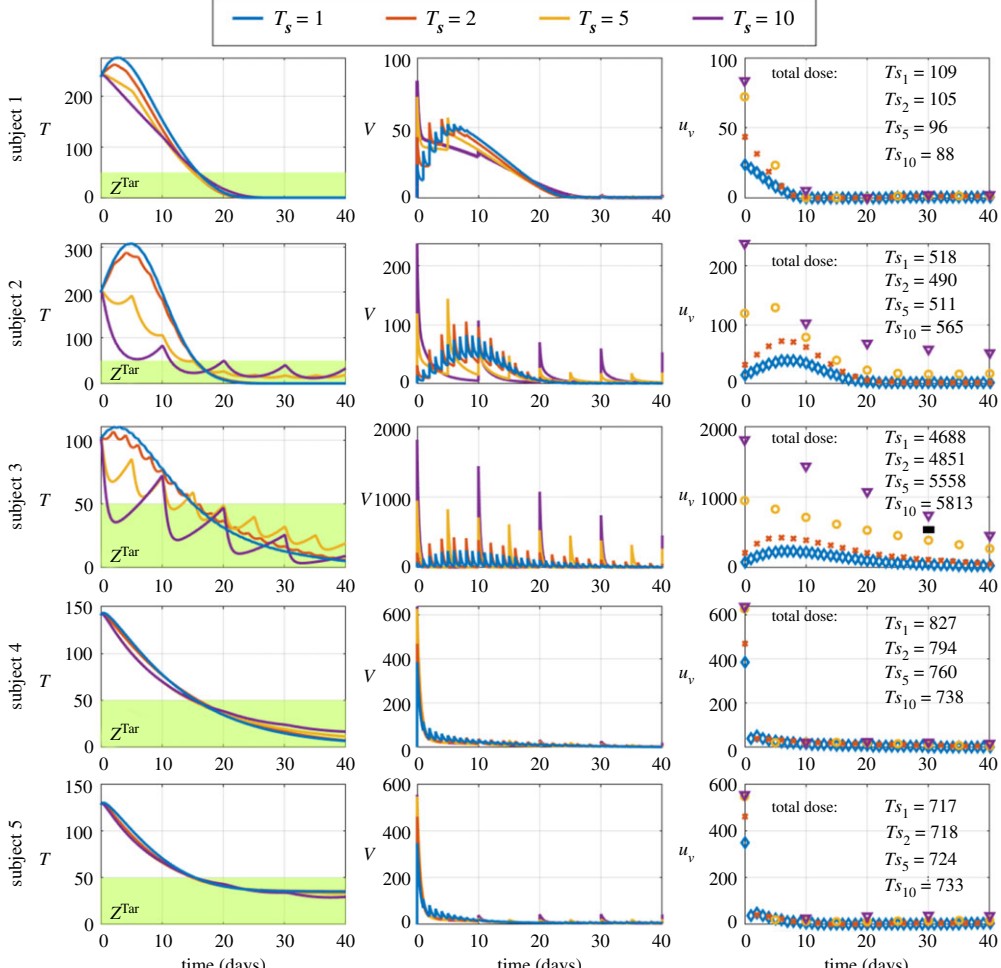

**Figure 4.** Feedback control delivers successful oncolytic virus therapy with different injection schedules. Oncolytic virus therapy is performed using feedback control to tune viral injections. $Ts_i$, $i = 1, 2, 5, 10$, refers to viral injections made every $i$ day. The blue lines and diamonds refer to $Ts_1$. The red lines and crosses refer to $Ts_2$. The orange lines and circles refer to $Ts_5$. The magenta lines and triangles refer to $Ts_{10}$. Each row shows the outcomes of therapies in a given subject. The left column shows the time course of the total number of tumour cells. The green region and $Z^{Tar}$ refer to the therapeutic zone. The middle column shows the time course of the viral loads *in vivo*. The right column shows the time course of viral injections.

injections; see figure 4. When the time between injections increases, the total viral load administered tends to decrease in subjects 1, 2 and 4, suggesting that lower dose with higher frequency is as effective as higher dose with lower frequency; see figure 4. Together, these results suggest that oncolytic virus therapy benefits from feedback control to tune viral injection appropriately for different schedules to achieve and sustain tumour regression.

Additionally, since different time constraints on tumour regression may apply for different patients, we investigated the impact of changes in the timings of tumour regression in each subject. We used the personalized therapy with one injection every 2 days as nominal response in each subject, and we compared the outcomes of tumour regression with a slower and faster time to enter the therapeutic zone. When the time to enter the therapeutic zone increases, the individual doses and the viral loads tend to be low at the beginning of therapy; see figure 5. Therefore, the tumour increases to higher levels at the beginning of therapy, when the time to enter the therapeutic zone increases; see figure 5. Subsequently, slow tumour regression leads to high viral loads and high doses during therapy in subjects 1, 2 and 3 to compensate for the high peak of tumour cells; see figure 5. Slow tumour regression leads to low viral loads and low doses during therapy in subjects 4 and 5. When tumour regression becomes slow, the total dose increases in subjects 1, 2 and 3, suggesting a negative correlation between the total doses and the speed of tumour regression to overcome the high peaks of the tumours; see figure 5. When tumour regression becomes slow, the total dose decreases in subjects 4 and 5, suggesting a positive correlation between the total doses and the speed of tumour regression;

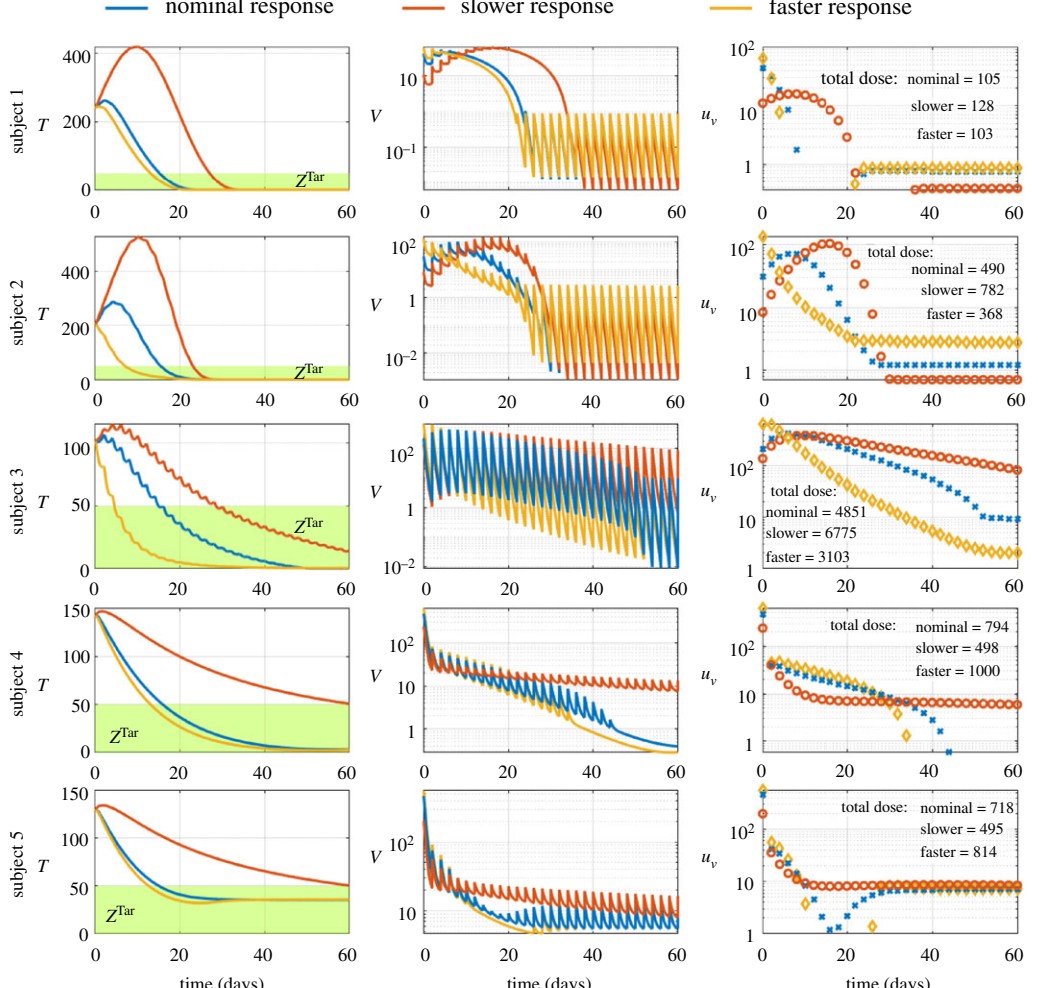

**Figure 5.** Feedback control delivers successful oncolytic virus therapy with different speed of tumour regression. The blue lines and blue crosses refer to the nominal response. The red lines and red circles refer to the slower response. The orange lines and orange diamonds refer to the faster response. Each row shows the outcomes of therapies in a given subject. The left column shows the time course of the total number of tumour cells. The green region and $Z^{Tar}$ refer to the therapeutic zone. The middle column shows the time course of the viral loads *in vivo*. The right column shows the time course of viral injections.

see figure 5. Thus, changes in the time course of tumour regression have different implications due to parameter variations between subjects. Collectively, these results suggest that oncolytic virus therapy benefits from feedback control to achieve and sustain tumour regression under different constraints.

## 3.5. Oncolytic virus therapy benefits from feedback control to formulate alternative dosages of viral injections

Since changes in the time course of viral injections could be beneficial to satisfy some preferences or constraints, we investigated whether similar tumour regression could be obtained by therapies which share similar total dose but differ in their time course of injections. We considered a schedule of one injection every two days and we compared the following two therapies which have similar total doses: (i) the time course of viral injections is dictated by the feedback control strategy which tunes the dosages of viral injections; and (ii) the time course of viral injections is dictated by an alternative protocol which changes the dosages in a staircase manner to facilitate therapy in practice; see Methods. Both therapies deliver very similar tumour regression in all subjects; the total number of tumour cells enter the therapeutic zone at the same time; see figure 6. Both therapies also have similar viral loads *in vivo* despite difference in the time course of viral injections; see figure 6. Together, these results suggest

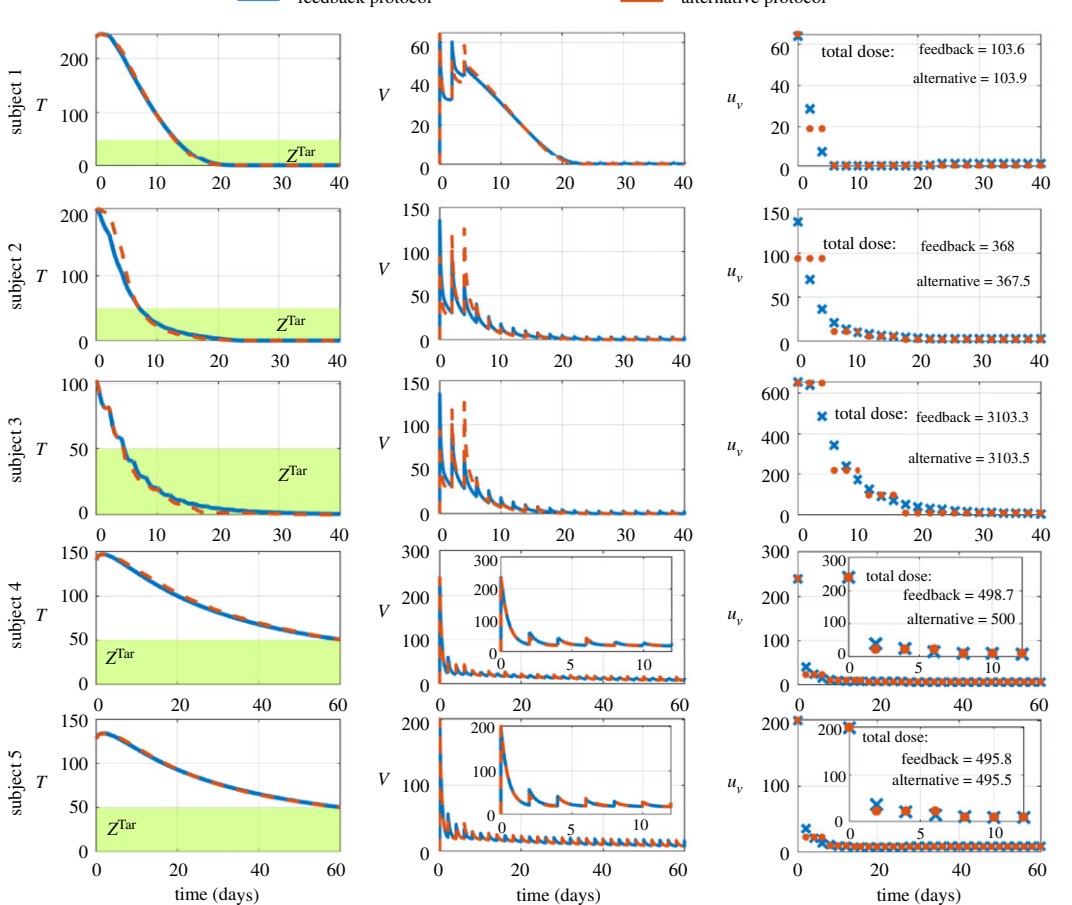

**Figure 6.** Robustness to changes in individual dosages of viral injections. Blue lines and blue crosses refer to the feedback protocol. Red lines and red circles refer to alternative protocol. Left column shows the time course of the total number of tumour cells. The middle column shows the time course of the viral loads *in vivo*. The right column shows the time course of viral injections.

that the efficacy of the therapy is robust to changes in the individual doses of viral injections as long as total dose reaches the required amount to achieve therapeutic objectives. Thus, applying a feedback control strategy is beneficial to determine the total dose to satisfy therapeutic objectives.

We also assessed the outcomes of common dosages for viral injections for all subjects. We compared personalized therapies with fast tumour regression to a common low dosage and a common high dosage. The common low dosage is the personalized therapy of subject 1 with the total dose of 105. The common high dosage is the personalized therapy of subject 4 with the total dose of 498. Tumour regression differs widely among subjects, suggesting that both dosages do not overcome variations in biological rates between subjects; see figure 7. The common low dosage fails to achieve tumour regression in subjects 2 to 5; see figure 7. By contrast, the common high dosage achieves tumour regression in all subjects, except subject 3, suggesting that the common dosage needs to be high enough to achieve tumour regression in each subject; see figure 7. The common high dosage tends to output a faster tumour regression in subjects 1 and 2, and slower tumour regression in subjects 4 and 5, suggesting that tumour regression becomes faster due to higher doses; see figure 7. The common high dosage misses the opportunity to achieve faster tumour regression with lower total doses in subjects 1 and 2, and may put these subjects at unnecessary risk of toxicity. As expected, the common high dosage delivers higher viral loads at the beginning than the common low dosage; see figure 7. However, viral loads decrease faster with the common high dosage than with the common low dosage because the common high dosage eliminates tumour cells faster; see figure 7. This suggests potential trade-offs between dosages, tumour regression and toxicity in the formulation of common dosages. Together, these results suggest that variations in biological rates between subjects may not allow efficient dosages in some subjects to bear desirable outcomes in others, thus motivating the formulation of personalized and robust strategies for oncolytic virus therapy.

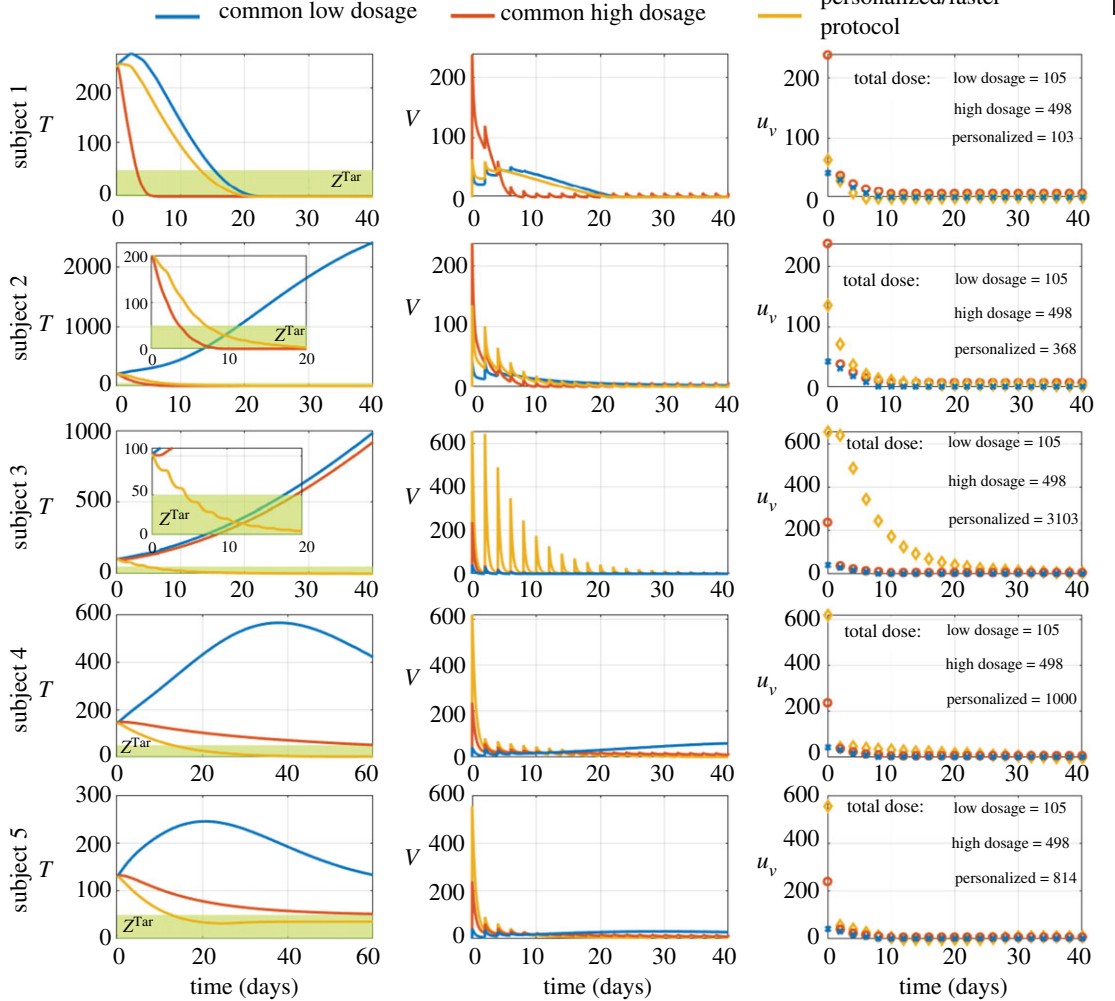

**Figure 7.** Advantages and disadvantages of common dosages of viral injections for all subjects. The blue lines and blue crosses refer to the common low dosage. This is the personalized therapy of subject 1 with the total dose of 110. The red lines and red circles refer to the common high dosage. This is the personalized therapy of subject 4 with the total dose of 498. The orange lines and orange diamonds refer to the personalized therapy with fast tumour regression of the corresponding subject. Each row shows the outcomes of therapies in a given subject. The left column shows the time course of the total number of tumour cells. The green region and $Z^{Tar}$ refer to the therapeutic zone. The middle column shows the time course of the viral loads *in vivo*. The right column shows the time course of viral injections.

## 3.6. Feedback control ensures that tumour regression exhibit some robustness to variations in biological rates

Since variations in biological rates may impair the outcomes of oncolytic virus therapy, we investigated whether tumour regression is robust to some variations in biological rates when oncolytic virus therapy is performed using the proposed feedback control strategy. We conducted a sensitivity analysis to identify parameters which mainly influence the behaviour of the system (2.1); see Methods. The sensitivity analysis indicates that variations in $r$, $\beta$, $d_v$ or $\alpha$ have greater impacts on behaviour of the system than variations in $L$, and $d_I$; see figure 8 and refer to table 3. The parameters $r$, $\beta$, $d_v$ and $\alpha$ are the most influential on the output $T$. Since the sign of sensitivity function $S_{\xi,p}$ is positive for the parameters $r$ and $d_v$, this predicts that the parameters $r$ and $d_v$ have a direct effect on the number of tumour cells; see figure 8. By contrast, the parameters $\beta$ and $\alpha$ have an indirect effect because $S_{\xi,\beta}$ and $S_{\xi,\alpha}$ are negative; see figure 8. These results suggest that large variations in the biological rates $r$, $\beta$, $d_v$ or $\alpha$ are likely to disturb the ability of the proposed feedback control strategy to achieve tumour regression during oncolytic virus therapy.

Based on these results, parametric variations are applied on these four parameters to evaluate the performance of the controller and the impact on tumour regression. We tested variations up to $\pm 30\%$

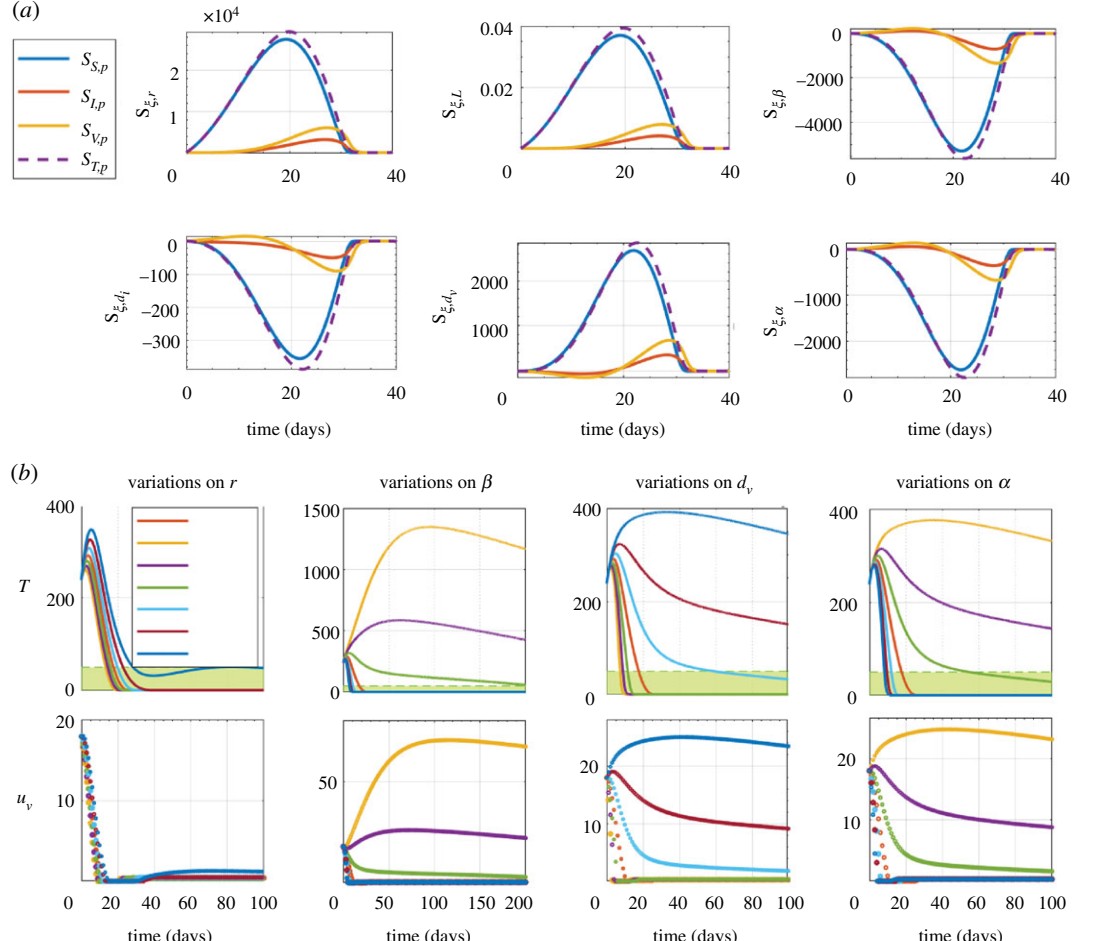

**Figure 8.** (a) The sensitivity function $S_{\xi,p}$ is plotted for all parameters. The parameters are defined in table 3. (b) System evolution under the state feedback controller when there are parameter variations.

**Table 3.** Description of the parameters of the model (2.1).

| symbol | meaning | unit |
| --- | --- | --- |
| $r$ | tumour growth rate | day$^{-1}$ |
| $L$ | carrying capacity | cells $\times 10^6$ |
| $\beta$ | infection rate of tumour cells | day$^{-1}$ |
| $d_I$ | death rate of infected tumour cells | day$^{-1}$ |
| $d_V$ | viral decay rate | day$^{-1}$ |
| $\alpha$ | viral burst size | virus $\times 10^9$ |

for each parameter. The results are consistent with the ones obtained from the sensitivity analysis; see figure 8. The control objective (to reach the target zone in less than 60 days) is achieved for variations up to $\pm 10\%$ in almost all parameters; see figure 8. For smaller variations than the nominal value of parameters $\beta$ and $\alpha$, and greater variations than the nominal value in parameter $d_v$, a slower response of the control system is observed. This leads to higher levels of tumour cells. In these situations, given the feedback of the system the amount of virus to be injected tends to increase, but the control action results are slower than the one of the nominal situation, and therefore so is the elimination of tumour cells. Collectively, these results suggest that the proposed feedback control strategy allows tumour regression during oncolytic virus therapy to exhibit some robustness to uncertainty in biological rates. Accurate estimation of the biological rates remains essential for better efficiency.

# 4. Discussions

Our findings build on previous studies of oncolytic virus therapy by applying control theory to deliver new insights and efficient protocols. A number of experimental studies have been conducted to test the concept of oncolytic virus therapy [5,6]. These studies presented both successful and unsuccessful cases of tumour regression [5,6]. These experiments can be regarded as input–output tests to check whether a given protocol for viral injection achieves tumour regression. By contrast, it is typical in the domain of control engineering to start analysing a system by performing controllability or accessibility tests on a suitable model, to prove that the chosen control input can indeed change the states of interest to the desired levels in finite time [16,24,31]. The control analysis of the model suggests that the tumour is controllable after at least the third injection. This analytical result is consistent with the experimental findings in [5], because the experiments achieved tumour regression in some mice using three viral injections. Although controllability and accessibility tests do not indicate how to design a suitable controller, accessibility and controllability tests prove analytically that it is possible to do so. This suggests that oncolytic virus therapy might benefit from accessibility and controllability tests to perform mathematical proof-of-concepts for different virus and protocols.

In this study, toxicity was not included as a constraint in the design of the controller and in the subsequent dosages of viral injections. The toxicity tests in [5,32] showed that when the surface of the oncolytic adenovirus is shielded with a biocompatible polymer such as polyethylene glycol (PEG), the virus therapy causes no hepatic damage and negligible liver toxicity. Future mathematical and experimental studies are required to improve understanding of the relationship between toxicity, doses of viral injections and viral load *in vivo*. These insights would be useful to inform the design of virus therapies and control strategies to ensure tumour regression with low toxicity.

Since the dose at a given day tends to increase when the time between injections increases, safe high-dose injections could be delivered to grant injection-free periods to patients and clinicians. The results here presented suggest that the outcomes of therapy mainly depend on the total amount of injected virus rather than the scheduling and time course of viral injections. This feature provides flexibility in the design of protocols. Thus, protocols could be adjusted to satisfy different preferences and constraints such as the number of follow visits, the posology of viral injections, or toxicity.

A given dosage of viral injection may perform differently in different subjects due to difference between the biological rates of each subject. Some subjects exhibit a beneficial property because the total dose and frequency of viral injections decrease up to some extent without compromising treatment outcomes. When biological rates are known, personalized protocols could be formulated to take advantage of patient-specific dynamics to improve the outcomes of therapy. When biological rates are uncertain, it is sensible to design therapies with robust control strategies to achieve desired outcomes despite uncertainties.

When high doses are injected at the beginning of the therapy in any subject, this reduces the peak of the total number of tumour cells and the total doses. However, this scenario raises concerns about toxicity due to increased viral loads *in vivo* and the eventual high cost of viral injections. The results presented suggest trade-offs between the time to reach the therapeutic zone, the maximum tumour size expected during therapy, the viral loads *in vivo*, and available doses of viral injections. These findings are consistent with the ones in [11,14].

By tuning the control strategy to obtain only three injections that bring the tumour cells to the target, the degrees of freedom are reduced in the design to decide the time of entry to the area and the maximum of allowed tumour cells. Therefore, there is no guarantee that all patients enter the zone at the same time. This is because there is nothing in the formulation of the feedback controller that allows to handle the behaviour of the control action, so a trial-and-error method is required. In control strategies such as model predictive control, a predetermined form of the control input could be obtained by imposing constraints such as $u(i) = u_{ss}$, for $i > = 4$.

The results of the sensitivity and robustness analysis are consistent with findings in [11,14]. In particular, the infection rate $\beta$ is crucial for the outcomes of therapy. As parameter $\beta$ becomes smaller, a higher control action is required to adequately regulate tumour cells. When the actual value of $\beta$ is 10% smaller than the identified value, the system reaches the target after 200 days (data not shown). This shows the importance of a good identification of parameter $\beta$.

Robustness to parameter variations can be improved by increasing the magnitude of the control input, i.e viral doses [16,25,29,30]. Since high viral doses may increase toxicity, the robustness of therapies may be limited by safety constraints on viral doses. On the one hand, this promotes the

development and application of optimal control strategies to minimize toxicity in the patient. On the other hand, this promotes the development and application of advanced robust control strategies such that higher percentages of parametric variations are adequately compensated.

In this study, viral injections are made at discrete time. However, if a continuous supply of virus is administered to the patient, for example via the blood or the respiratory tract, oncolytic virus therapy could benefit from advanced continuous time control strategies to regulate the system in different state values and sustain tumour regression in the presence of biological uncertainty.

In conclusion, this paper presents the first scheme to perform oncolytic virus therapy using impulsive control theory. Although we note caveats associated with experimental testing of the proposed protocols, these could be alleviated by including constraints related to viral doses and toxicity. Together, the analysis and results here presented contribute to knowledge on oncolytic virus therapy and open new avenues based on control theory to develop effective therapies. In future works, it is envisioned experimental trials to verify the analytical results developed here.

Ethics. This heading is not relevant for this work.

Data accessibility. The datasets supporting this article have been uploaded as part of the electronic supplementary material.

Authors' contributions. A.J.N.A. conceived the study, described the nonlinear model in §2 and carried out the identification of parameters illustrated in the appendix. Also, he helped draft the manuscript. P.S.R. contributed with the theoretical framework of the study, assessing the equilibria of the model, controllability tests, impulsive control strategy and robustness analysis. M.F.V.-T. developed the code for the analysis and results, and the artwork of the paper. Also, she collaborated with the theoretical part regarding the different proposals for the treatment using impulsive feedback. All authors gave final approval for publication and agreed to be accountable for all aspects of the work in ensuring that questions related to the accuracy or integrity of any part of the work are appropriately investigated and resolved.

Competing interests. The authors declare no competing interests.

Funding. This work was supported by the Australian Research Council via the Discovery Project grant reference number DP180101512.

Acknowledgements. Anet dearly thanks Adrianne L. Jenner and Adelle Coaster.

# Appendix A. Methods

## A.1. Initial conditions and parameter values of the model

We considered experiments in [5] which tested treatment of breast cancer in mice by an oncolytic PEG-modified adenovirus conjugated with herceptin (Ad-PEG-HER). We used a nonlinear least-squares method i.e. the function lsqnonlin() in Matlab R2019a to fit the model (2.1) to times series of the total number of tumour cells from [5] as in [14]. As expected, the model fits well the data as in [14]; see figure 9. The corresponding initial conditions and biological rates are given in table 1; and the corresponding goodness of fit statistics are given in table 4.

## A.2. Equilibria of the model

We investigated the equilibria of the model (2.1) because this is important, as it gives an idea of the points at which the system (2.1) can be forced, and we can be certain that the system will remain there. The equilibria of the model (2.1) are also used to linearize the model (2.1) to perform controllability tests and design linear control strategies.

To determine the equilibria, we considered two cases. The first case corresponds to the autonomous system without viral injection, i.e. $u(t) = 0$; the second case to the control system with viral injection i.e. $u(t) \neq 0$.

When $u(t) = 0$, the model (2.1) has three equilibria

$$Eq_h = (0\ 0\ 0;\ 0), \tag{A 1}$$

$$Eq_e = (L\ 0\ 0;\ 0) \tag{A 2}$$

and

$$Eq_v = (S_{ss}\ I_{ss}\ V_{ss};\ 0), \tag{A 3}$$

where $Eq_h$ is the healthy equilibrium. The second one is characterized by $u_{vss} = 0$, $V_{ss} = 0$, $I_{ss} = 0$ and $S_{ss} = L$, so the maximum number of tumour cells is reached. This equilibrium, $Eq_e$, is considered as the 'endemic' equilibrium. The third equilibrium $Eq_v$ is also an endemic point with all values different

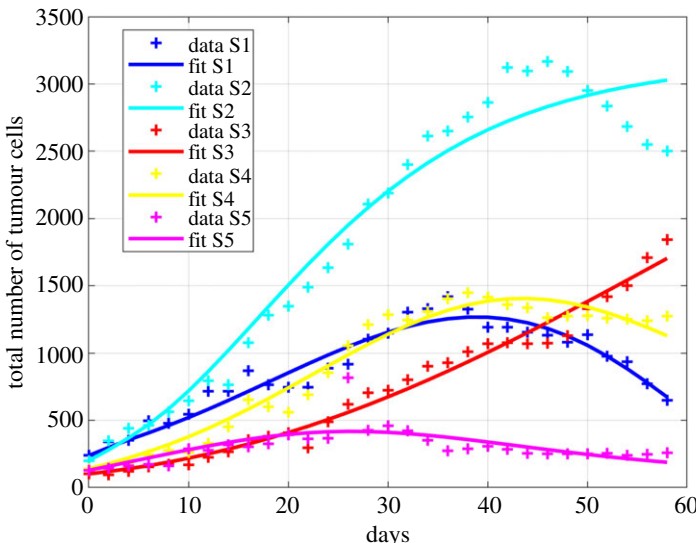

**Figure 9.** Model–data fits for Ad-PEG-HER. The crosses are the measurements of the total number of tumour cells. The solid lines are the time course of the total number of tumour cells from the model (2.1). Each colour is assigned to a given subject $Si$ for $i = 1, 2, 3, 4, 5$.

**Table 4.** Goodness of fit statistics for the model (2.1) for Ad-PEG-HER.

| subject | $R^2$ | Pearson correlation coefficient | $p$-value of the Pearson correlation |
|---|---|---|---|
| S1 | 0.93 | 0.96 | $4.3872 \times 10^{-18}$ |
| S2 | 0.95 | 0.97 | $2.4499 \times 10^{-20}$ |
| S3 | 0.97 | 0.98 | $1.0721 \times 10^{-24}$ |
| S4 | 0.96 | 0.98 | $2.1155 \times 10^{-22}$ |
| S5 | 0.54 | 0.74 | $2.8960 \times 10^{-6}$ |

from zero. It is described by $u_{vss} = 0$, $I_{ss} = (d_v/\alpha d_I)V_{ss} = c_1 V_{ss}$, $S_{ss} = (d_v^2/(\beta\alpha - d_v)\alpha d_I)V_{ss} = c_2 V_{ss}$, and $V_{ss} = (L/c_2)\, \mathrm{e}^{-(\beta/r)1/(c_1+c_2)}$.

When $u(t) = u_{vss} \neq 0$, additional equilibria of the form (A 3) are generated

$$\left.\begin{array}{l} S_{ss} = f(S, u_{vss}), \\ I_{ss} = \frac{r}{d_I}\ln\left(\frac{L}{S_{ss}}\right)S_{ss} \\ V_{ss} = \frac{ad_I I_{ss} + u_{vss}}{d_v}, \end{array}\right\} \tag{A 4}$$

and

where $f(S, u_{vss})$ is the solution of equation: $rd_I \ln(L/S)S + (rd_I^2/S)(\ln(L/S) - 1)\ln(L/S)S - u_{vss} = 0$.

Since the stability of each equilibrium was analysed in [11], we build on that work by investigating the controllability of the system (2.1) linearized at each equilibrium and the accessibility of the nonlinear system in the next sections.

## A.3. Impulsive systems

Impulsive control theory is used because of the form in which the virus enters the body; viral injections represent impulses which aim to achieve tumour regression. Consequently, the model for oncolytic virus therapy (2.1) can be rewritten as an impulsive control system of the form

$$\begin{array}{rcll} \dot{\xi}(t) &=& f(\xi(t)), & \xi(0) = \xi_0, \ t \neq \tau_k, \\ \xi(\tau_k^+) &=& \xi(\tau_k) + g(\xi(\tau_k))u(\tau_k), & t = \tau_k, \quad k \in \mathbb{N}, \\ Z(\tau_k) &=& h(\xi(\tau_k)), & t = \tau_k, \quad k \in \mathbb{N}, \end{array} \tag{A 5}$$

where $g(\xi(\tau_k))$ denotes the effect of viral injections at each impulsive time $\tau_k$, $k = 1, 2, \ldots$ over the state $\xi$; and $Z(\tau_k) = h(\xi(\tau_k))$ is the output function. Similarly, the linearized ICS around an equilibrium point is

described by

$$
\begin{aligned}
\dot{x}(t) &= Ax(t), & x(0) = x_0, \ t \neq \tau_k, \\
x(\tau_k^+) &= x(\tau_k) + Bu(\tau_k), & t = \tau_k, \quad k \in \mathbb{N}, \\
y(\tau_k) &= Cx(\tau_k), & t = \tau_k, \quad k \in \mathbb{N}.
\end{aligned}
\tag{A 6}
$$

Note that, in (A 5) and (A 6), the respective differential equation represents the autonomous response, and the algebraic equation represents the discontinuities or 'jumps' caused by the control action. With a view to designing linear control strategies, it is useful to discretize the ICS (A 6) at times $\tau_k$ and $\tau_k^+$. This is done by considering the effect of the impulsive form of input $u$ over state $x$. Then, system (A 6) can be characterized by means of two underlying discrete-time subsystems (UDS) as [15]

$$
x^\bullet(k+1) = A^\bullet x^\bullet(k) + B^\bullet u^\bullet(k), \quad x^\bullet(0) = x(\tau_0)
\tag{A 7a}
$$

and

$$
x^\circ(k+1) = A^\circ x^\circ(k) + B^\circ u^\circ(k), \quad x^\circ(k) = x(\tau_0^+),
\tag{A 7b}
$$

where $x^\bullet(k)$ and $x^\circ(k)$ denote the deviation variables $x(\tau_k)$ and $x(\tau_k^+)$ of the linearized system. The corresponding matrices are defined by $A^\bullet \triangleq A^\circ \triangleq e^{AT_s}$, $B^\bullet \triangleq e^{AT_s}B$ and $B^\circ \triangleq B$. For the input, it is fulfilled that $u^\bullet(k) = u^\circ(k+1)$, for $k \in \mathbb{N}$. The fixed sampling time $T_s$ is defined as $T_s = \tau_{k+1} - \tau_k$. A complete characterization of equilibrium and target sets for both UDS and subsequently for the ICS (A 6) is presented in [33].

## A.4. Controllability tests

The notion of controllability determines whether a control law can steer a linear system from any initial state to other (final) state in finite time. We analysed the controllability of model (2.1) because it is a fundamental property to study before designing a linear control strategy for oncolytic virus therapy. We performed controllability tests on the model (2.1) linearized at its equilibria.

The nonlinear model (2.1) around an equilibrium point can be written as (A 6), where $x = \xi - \xi_{ss}$, $A = \partial f(\xi, u)/\partial \xi \mid_{Eq}$, $B = \partial f(\xi, u)/\partial u \mid_{Eq}$ and $C = \partial h(\xi)/\partial \xi \mid_{Eq}$, with $h(\xi) = S + I$ the output equation. These matrices are called the Jacobians of the system. The Jacobian matrix $A$ is given by

$$
A = \begin{pmatrix}
-r + r\ln\left(\frac{L}{S}\right) - \beta\frac{IV}{(I+S)^2} & \beta\frac{VS^2}{(I+S)^2} & -\beta\frac{S}{I+S} \\
\beta\frac{IV}{(I+S)^2} & -\beta\frac{IV}{(I+S)^2} - d_I & \beta\frac{S}{I+S} \\
0 & \alpha d_I & -d_V
\end{pmatrix}.
\tag{A 8}
$$

Note that the matrix $A$ will change depending on the selected equilibrium, but matrices $B$ and $C$ are the same for any equilibrium point, holding the expressions $B = \begin{pmatrix} 0 \\ 0 \\ 1 \end{pmatrix}$, and $C = (1\ 1\ 0)$.

The controllability depends on the system structure, i.e. on the pair $(A, B)$. When the controllability matrix

$$
\mathcal{C} = (B \ AB \ A^2 B)
$$

has full rank, the impulsive linear system is controllable [25]. The linear model of (2.1) is not controllable at the equilibrium $Eq_h$ (A 1). This is expected because all variables are zero. When the model (2.1) is linearized at the endemic equilibrium $Eq_e$ (A 2), the Jacobian matrix is

$$
A_1 = \begin{pmatrix}
-r & 0 & -\beta \\
0 & -d_I & \beta \\
0 & \alpha d_I & -d_V
\end{pmatrix},
\tag{A 9}
$$

and the corresponding controllability matrix is

$$
\mathcal{C} = \begin{pmatrix}
0 & -\beta & \beta(r + d_V) \\
0 & \beta & -\beta(d_I + d_V) \\
1 & -d_V & \alpha\beta d_I + d_V^2
\end{pmatrix}.
\tag{A 10}
$$

The determinant of $\mathcal{C}$ is $\det(\mathcal{C}) = \beta^2(r - d_I)$, which is different from zero if $r \neq d_I$. In this case, $\text{Rank}(\mathcal{C}) = 3$. Therefore, the model (2.1) is controllable around the equilibrium $Eq_e$ (A 2).

When the model (2.1) is linearized at the endemic equilibrium $Eq_V$ (A 3), the Jacobians matrix is

$$A_2 = \begin{pmatrix} r\left(\ln\left(\frac{L}{c_2 V_{ss}}\right) - 1\right) - \beta\frac{c_1}{(c_1+c_2)^2} & \beta\frac{c_2}{(c_1+c_2)^2} & -\beta\frac{c_2}{c_1+c_2} \\ \beta\frac{c_1}{(c_1+c_2)^2} & -d_I - \beta\frac{c_2}{(c_1+c_2)^2} & \beta\frac{c_2}{c_1+c_2} \\ 0 & \alpha d_I & -d_V \end{pmatrix}. \tag{A 11}$$

The corresponding controllability matrix

$$\mathcal{C} = \begin{pmatrix} 0 & -\beta\frac{c_2}{c_1+c_2} & \beta\frac{c_2}{c_1+c_2}\left(\frac{\beta}{c_1+c_2} + d_i + d_V + r - r\ln\left(\frac{L}{c_2 V_{ss}}\right)\right) \\ 0 & \beta\frac{c_2}{c_1+c_2} & -\beta\frac{c_2}{c_1+c_2}\left(\frac{\beta}{c_1+c_2} + d_i + d_V\right) \\ 1 & -d_V & \alpha\beta d_I d_V^2 \frac{c_2}{c_1+c_2} \end{pmatrix} \tag{A 12}$$

has full rank but subject to $\det(\mathcal{C}) \neq 0$, which leads to

$$\det(\mathcal{C}) = \beta^2 \frac{c_2^2}{(c_1+c_2)^2}\left(d_i - r + r\ln\left(\frac{K}{c_2 V_{ss}}\right)\right) \rightarrow d_i \neq r - r\ln\left(\frac{K}{c_2 V_{ss}}\right). \tag{A 13}$$

Therefore, the model (2.1) is controllable around the equilibrium $Eq_V$. Thus, it is sensible to design control strategies for the model (2.1) around the equilibrium points $Eq_e$ and $Eq_V$.

Furthermore, we used results on the controllability of linear impulsive control systems (see [25,29]) to determine the minimum number of impulses, i.e. injections required to control the tumour from any initial level to a desired level around the equilibrium points $Eq_e$ and $Eq_V$ of the model (2.1). The control analysis in [25] found that the minimum number of impulses is equal to the controllability index of the pair $(A, B)$ for the linear impulsive control system. The controllability index refers to the number of linearly independent columns associated with the control input [30,34]. Since the model (2.1) is controllable around the equilibrium points $Eq_e$ and $Eq_V$, the controllability index is $Rank[\mathcal{C}] = Rank[B, AB, A^2B] = n = 3$ for the model (2.1) linearized around the equilibrium points $Eq_e$ and $Eq_V$.

## A.5. Accessibility characterization of oncolytic virus dynamics

To generalize the analysis of the controllability property to the nonlinear system we investigate the accessibility of model (2.1). First, with the purpose of showing that the output $h(\xi) = T = S + I$ is not an autonomous element for the system, and therefore, at least three injections are required to control the system, we compute the relative degree of the system. By applying the condition (ii) of proposition 2 in [25], we get

$$\left.\begin{aligned} \langle dh(\xi), g(\xi)\rangle &= \frac{\partial h(\xi)}{\partial \xi} g(\xi) = (1 \quad 1 \quad 0)(0 \quad 0 \quad 1)' = 0, \\ \langle dL_f h(\xi), g(\xi)\rangle &= L_g L_f h(\xi) = \left(r(\ln(\tfrac{L}{S}) - 1) \quad -d_I \quad 0\right)(0 \quad 0 \quad 1)' = 0 \\ \langle dL_f^2 h(\xi), g(\xi)\rangle &= L_g L_f^2 h(\xi) = -\beta\frac{S}{S+I}\left(d_I + r(\ln\left(\frac{L}{S}\right) - 1)\right). \end{aligned}\right\} \tag{A 14}$$

and

From equation (A 14), we can conclude that the impulse relative degree is $d^0(y) = 3$, as a matter of fact $\langle dL_f^2 h(\xi), g(\xi)\rangle \neq 0$ for any $S > 0$, and $d_I \neq r(\ln(\tfrac{L}{S}) - 1)$, i.e the impulsive input affects the output $h(\xi) = S + I$ after at least the 3-th impulse, therefore $h(\xi)$ is not an autonomous element of the system. However, this adds a constraint in the reachable space, which becomes the strictly positive subspace of $\mathbb{R}^3$. Now, From theorem 2 in [25], we obtain

$$\text{ad}_f g = \frac{\partial g}{\partial \xi} f - \frac{\partial f}{\partial \xi} g = \begin{pmatrix} \beta\frac{S}{S+I} \\ -\beta\frac{S}{S+I} \\ d_v \end{pmatrix}$$

and

$$\text{ad}_f^2 g = \frac{\partial \text{ad}_f g}{\partial \xi} f - \frac{\partial f}{\partial \xi} \text{ad}_f g = \begin{pmatrix} a_1(S, I) \\ a_2(S, I) \\ d_v^2 + \beta\alpha d_I \frac{S}{S+I} \end{pmatrix},$$

where

$$a_1(S, I) = \left(\beta\left(Ip(S) + S\left(d_v S + (d_I + d_v)I - (S + I)\frac{dp}{dS}\right)\right)\right)/(S + I)^2,$$

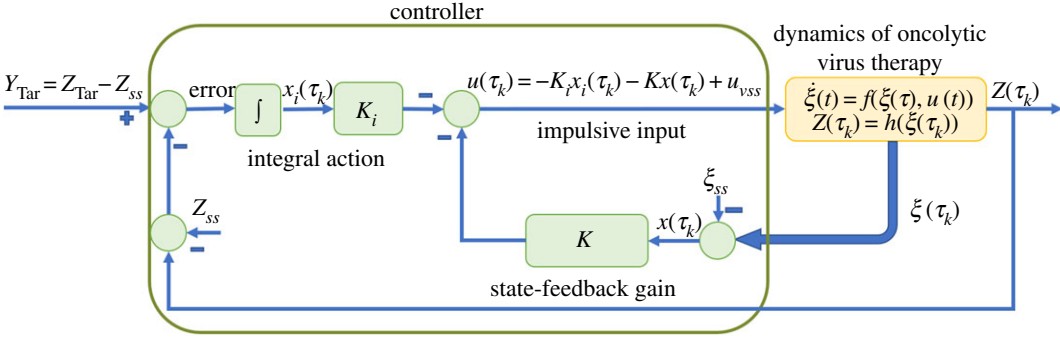

**Figure 10.** Impulsive control strategy for oncolytic virus therapy. The description of each symbol can be seen in table 5.

**Table 5.** Description of each variable in figure 10.

| symbol | meaning |
|---|---|
| $\xi$ | state of the nonlinear system |
| $\xi_{ss}$ | equilibrium state of the nonlinear system |
| $Z$ | output of the nonlinear system |
| $Z_{ss}$ | equilibrium point of system output |
| $x$ | linearized state around the equilibrium point $\xi_{ss}$ |
| $Y_{Tar}$ | target for the linear system |
| $Z_{Tar}$ | target for the nonlinear system |
| $x_i$ | integrate state |
| $K_i$ | gain of the integral state |
| $K$ | state feedback gain |
| $u$ | impulsive control input |
| $u_{vss}$ | impulse equilibrium input |

and

$$a_2(S, I) = -((\beta(S(d_v(S + I) + d_I(S + 2I)) + Ip(S))/(S + I)^2).$$

The determinant is

$$\det(g, \mathrm{ad}_f g, \mathrm{ad}_f^2 g) = -\beta^2 \frac{S^2}{(S + I)^2}\left(d_I + r(\ln\left(\frac{L}{S}\right) - 1)\right). \tag{A 15}$$

Equation (A 15) shows that the dimension of $\dim\overline{\{g, \mathrm{ad}_f g, \mathrm{ad}_f^2 g\}}$ is equal to 3 for any $S > 0$ and $S \neq Le^{-(1+d_i/r)}$. As a result, the strictly positive subspace $\mathbb{R}_+^3$ is accessible.

## A.6. State feedback control

The purpose of using a control strategy is to achieve an alteration over the dynamical system such that it behaves as a different system that accomplishes desired objectives. We used an impulsive control strategy to alter the time course of the tumour such that tumour dynamics during therapy accomplish to reduce the density of tumour cells $T = S + I$ below 50 cells within 60 days, i.e. to steer and maintain the system into the zone $Z^{Tar} = \{T : 0 \leq T \leq 50\}$, where $Z_{Tar} = 0$ is chosen as a target or set-point within the therapeutic zone. Viral injections are restricted to a maximum of one injection per day. We proposed a state feedback control strategy with an integral action which tunes the dosages of viral injections, to achieve tumour regression; see figure 10 and table 5.

As we define above, the target for the nonlinear system is the healthy equilibrium, i.e. $Z_{Tar} = 0$. However, we use a linearized system around a different equilibrium $\xi_{ss}$ near the healthy equilibrium to control the nonlinear system. Therefore, the target for the linear system is $Y_{Tar} = Z_{Tar} - h(\xi_{ss}) = -$

$h(\xi_{ss}) = -Z_{ss}$, which implies that the control problem for the nonlinear system is a regulation problem, while for the linear system is a tracking problem.

Based on this, the idea of the strategy consists of changing the poles of the system by moving the value of a feedback gain $K$; i.e. to alter the stability and transient properties of the system. However, gain matrix $K$ only meets the transient requirements. Therefore, an integral action is required to drive the system to the desired linear target at steady state. In addition, the integral action will account for enlarging the robustness of control due to parameter variations which often occur in practice. In this way, a robust asymptotic track of the reference will be ensured. To achieve this design, we will introduce the new state $\dot{x}_i = Y_{Tar} - Cx$. Thus, for linear systems, the control law is expressed by

$$u(t) = \kappa(\tilde{x}(t)) = -Kx(t) - K_i x_i = -\tilde{K}\tilde{x}(t), \tag{A 16}$$

where $\tilde{x} = (x \ \ x_i)'$ is the augmented state, and $\tilde{K} = (K \ \ K_i)$ is the augmented gain matrix. Remarking that viral injections $u$ only affects the system at instants $\tau_k$, $k = 0, 1, \ldots$, then the augmented impulsive state system results in

$$\left.\begin{aligned}
\dot{\tilde{x}}(t) &= \tilde{A}\tilde{x}(t) + B_r Y_{Tar}, \quad x(0) = x_0, \ t \neq \tau_k, \\
\tilde{x}(\tau_k^+) &= \tilde{x}(\tau_k) + \tilde{B}u(\tau_k), \quad t = \tau_k, \quad k \in \mathbb{N} \\
y(\tau_k) &= \tilde{C}\tilde{x}(\tau_k), \quad t = \tau_k, \quad k \in \mathbb{N},
\end{aligned}\right\} \tag{A 17}$$

and

where $\tilde{A} = \begin{pmatrix} A & 0 \\ -C & 0 \end{pmatrix}$, $\tilde{B} = \begin{pmatrix} B \\ 0 \end{pmatrix}$, $B_r = \begin{pmatrix} 0 \\ 1 \end{pmatrix}$, $\tilde{C} = (C \ 0)$. The design of the control strategy is based on the selection of gain matrices $K$ and $K_i$, by considering their effect on the UDS (A 7). To that end, any of the UDS can be used as shown next. According to the discretization, the control action is applied on $x^\bullet$. Then the closed-loop form of the first UDS results in

$$\begin{aligned}
\tilde{x}^\bullet(k+1) &= (\tilde{A}^\bullet - \tilde{B}^\bullet \tilde{K})\tilde{x}^\bullet(k) + B_r^\bullet Y_{Tar} \\
&= e^{\tilde{A}T_s}(I_n - \tilde{B}\tilde{K})\tilde{x}^\bullet(k) + B_r^\bullet Y_{Tar}.
\end{aligned} \tag{A 18}$$

Now, taking into account the relation $\tilde{x}(\tau_k^+) = (I_n - \tilde{B}\tilde{K})e^{\tilde{A}T_s}\tilde{x}(\tau_{k-1}^+)$, the closed-loop form of the second UDS is given by

$$\begin{aligned}
\tilde{x}^\circ(k+1) &= \tilde{A}^\circ \tilde{x}^\circ(k) + \tilde{B}^\circ(-\tilde{K}\tilde{x}^\bullet(k)) + B_r^\circ(k)Y_{Tar} \\
&= (I_n - \tilde{B}\tilde{K})e^{\tilde{A}T_s}\tilde{x}^\circ + B_r^\circ(k)Y_{Tar}, 
\end{aligned} \tag{A 19}$$

with $I_n$ the identity matrix of dimension $n$. Then, it can be proved that both equations have the same characteristic polynomial

$$\begin{aligned}
\det(\lambda I_n - e^{\tilde{A}T_s}(I_n - \tilde{B}\tilde{K})) &= \det(e^{\tilde{A}T_s}(e^{-\tilde{A}T_s}\lambda I_n - (I_n - \tilde{B}\tilde{K}))) \\
&= \det(e^{\tilde{A}T_s})det(e^{-\tilde{A}T_s}\lambda I_n - (I_n - \tilde{B}\tilde{K})) \\
&= \det((\lambda I_n e^{-\tilde{A}T_s} - (I_n - \tilde{B}\tilde{K}))e^{\tilde{A}T_s}) \\
&= \det(\lambda I_n - (I_n - \tilde{B}\tilde{K})e^{\tilde{A}T_s}),
\end{aligned}$$

and therefore both subsystems have the same eigenvalues $\lambda$. With this in mind, matrices $K$ and $K_i$ are selected such that the eigenvalues of closed-loop system $(\tilde{A}^\bullet - \tilde{B}^\bullet \tilde{K})$ or $(\tilde{A}^\circ - \tilde{B}^\circ \tilde{K})$ remain inside the unitary circle, to ensure that the therapeutic target is attractive for the system [15]. In addition, since the control input $u$ is applied to the nonlinear plant, the impulse equilibrium input $u_{vss}$ used to linearize the system must be added to the calculated input.

## A.7. Configuring the controller to achieve desired tumour regression

A state feedback control strategy requires the computation of an appropriate feedback gain to tune the dosage of viral injection such that the therapy reaches its objective. As the objective of oncolytic virus therapy is to reduce the total number of tumour cells, i.e. less than or equal to 50 cells in our context, we need to steer the system (2.1) near the origin. Therefore, we linearized the system (2.1) around an equilibrium point near to the origin. This therapeutic equilibrium is found by choosing a value for $S_{ss}$, by solving (A 4) and by checking that $S_{ss} + I_{ss} \leq 50$. We emphasized that it is sensible to use the

controllability test (§3.6) to confirm that the linearized system at this point is controllable before designing any control strategy. When the therapeutic equilibrium is controllable, we tuned the feedback gains by placing eigenvalues appropriately to stabilize the closed-loop system at the therapeutic equilibrium, and to ensure that tumour regression satisfies therapeutic objectives and constraints. This tuning process is done by considering that eigenvalues near the unitary circle are dominant over eigenvalues near the origin. As eigenvalues approach to the unitary circle the response of the system is slower. When the eigenvalues approach to the origin a faster response of the system is obtained. Then, the set of values for feedback gain matrix $\tilde{K}$ is calculated in Matlab using the function place() with the desired eigenvalues. Each configuration of the controller automatically computes the required dose at a given time of viral injection.

## A.8. Personalized viral injections with feedback control

We personalized oncolytic virus therapy by configuring the controller using the biological rates of given subject to personalize viral injections. We considered the numerical control analysis for subject S1 in table 1) as a representative case to compute personalized viral injections. The therapeutic equilibrium for this subject is $\xi_{ss} = (10\ 1.2741\ 2.5645)$ and $u_{vss} = 0.2575$. The controllability test (§3.6) confirmed that the linearized system is controllable at this therapeutic equilibrium. Next, we computed the numerical values of the elements of the feedback gain matrix $K$. These feedback gains are selected (i) to ensure the stable dynamics by placing the eigenvalues of the closed-loop system within the unit circle, and (ii) to ensure that the therapy meets its objective, i.e. to reduce the total number of cells below 50 in less than 60 days. Consequently, to reach the target zone on day 15 with a schedule of one injection every 2 days, the feedback gain matrix $\tilde{K}$ is: $\tilde{K} = (-0.1809\ 1.0619\ 0.0791\ 0.0002)$ for subject S1. Similarly, we repeated this procedure to personalized oncolytic virus therapy for subjects in table 1.

## A.9. Personalized therapy with the experimental schedule

The experimental therapies in [5] injected viruses only on days 0, 2 and 4. We used the proposed feedback control strategy to deliver one efficient and personalized injection on these days. Firstly, we formulated a personalized therapy with feedback control to reduce the tumour quickly to keep the total number of tumours close to zero during the follow-up period of 60 days. This is done to obtain an efficient total dose. Secondly, we obtained the personalized therapy with the experimental schedule by distributing the total dose in a staircase manner over three injections. The personalized therapy with the experimental schedule performs therapy with predefined doses, i.e. with no feedback control.

## A.10. Alternative staircase protocol

We formulated an alternative staircase protocol with the schedule of one injection every 2 days for a given subject using the reference total dose of that subject and the form of the input curve computed by the feedback controller. The reference total dose is the lowest total dose between the fast and slow responses exhibited by a given subject, during personalized therapies with feedback control and one injection every 2 days. We delivered doses in a staircase manner such that the total dose of this staircase protocol is very close to the reference total dose for each subject. Thus, the alternative staircase protocol is a personalized therapy with no feedback control.

## A.11. Robustness analysis

To expose the dependence of the system to the model parameters and their variations, a sensitivity analysis is performed. From this, we analyse which parameters are the most influential in the tumour growth. Assuming the identified parameters as the nominal value $p_n$ of the parameters $p$, the sensitivity function is defined as $S_{\xi_i, p_j} = \partial \xi_i / \partial p_j$. This function satisfies the following dynamical system:

$$\dot{S}_{\xi, p_j}(t) = \frac{\partial f(\xi(t), u, p_n)}{\partial \xi(t)} S_{\xi, p_j} + \frac{\partial f(\xi(t), u, p_n)}{\partial p_j}, \quad S_{\xi, p_j}(0) = 0, \quad \text{(A 20)}$$

where $S_{\xi, p_j}$ represents the sensitivity of the state $\xi$ with respect to parameter $p_j$. The vector of parameter is defined as $p = (r\ L\ \beta\ d_I\ d_v\ \alpha)$. The analysis of variations of the most influential parameters is done to retrieve information about the effect on the control performance when there is a plant-model

mismatch. The most influential parameters are found by comparing the magnitude of the sensitivity functions. Based on these results, parametric variations are applied to the most influential parameters to evaluate the robust performance of the proposed feedback control.

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
