## [Reviewer comments · Royal Society Open Science]

Review History

RSOS-200473.R0 (Original submission)

Review form: Reviewer 1

Is the manuscript scientifically sound in its present form?

No

Are the interpretations and conclusions justified by the results?

No

Is the language acceptable?

Yes

Do you have any ethical concerns with this paper?

No

Have you any concerns about statistical analyses in this paper?

No

Recommendation?

Reject

Comments to the Author(s)

The authors proposed an oncolytic virus therapy based on control theory. The model that was taken for the study was validated in a mouse model in previous published paper. Stability analysis was developed before. Unfortunately, I can not support the publication of the present study as it does not have a proper validation and/or comparison with any data.

Review form: Reviewer 2

Is the manuscript scientifically sound in its present form?

Yes

Are the interpretations and conclusions justified by the results?

Yes

Is the language acceptable?

Yes

Do you have any ethical concerns with this paper?

No

Have you any concerns about statistical analyses in this paper?

No

Recommendation?

Major revision is needed (please make suggestions in comments)

Comments to the Author(s)

See attachment (Appendix A).

Review form: Reviewer 3

Is the manuscript scientifically sound in its present form?

No

Are the interpretations and conclusions justified by the results?

Yes

Is the language acceptable?

No

Do you have any ethical concerns with this paper?

No

Have you any concerns about statistical analyses in this paper?

No

Recommendation?

Major revision is needed (please make suggestions in comments)

Comments to the Author(s)

The application of proper quantitative analyses like control theory to therapeutic viruses is a good approach and advances the field.

However the abstract makes it sound like this paper has achieved far more than it really has, and could thus misinform. The authors muddle what is shown on one particular model with general statements about curing and controlling tumours; they mustn't. First, it is presented as if three treatments has been generally demonstrated to be a minimum, rather than one particular model requiring three to fit a specific criterion. If the authors want to claim "Our results demonstrate that at least three viral injections are required" they need a general technique that can fairly compare one- and two- treatment strategies, or they need to make it clear that the `_model_` is controllable. Similarly, if they want to claim "Our results demonstrate that at least three viral injections are required to control and reduce the tumor..." they need real experiments. The correct description would be that the models suggest or imply it.

Decision letter (RSOS-200473.R0)

16-Apr-2020

Dear Dr Rivadeneira,

The editors assigned to your paper ("Oncolytic Virus Therapy Benefits from Control Theory") have now received comments from reviewers. We would like you to revise your paper in accordance with the referee and Associate Editor suggestions which can be found below (not including confidential reports to the Editor). Please note this decision does not guarantee eventual acceptance.

Please submit a copy of your revised paper before 09-May-2020. Please note that the revision deadline will expire at 00.00am on this date. If we do not hear from you within this time then it will be assumed that the paper has been withdrawn. In exceptional circumstances, extensions may be possible if agreed with the Editorial Office in advance. We do not allow multiple rounds of revision so we urge you to make every effort to fully address all of the comments at this stage. If deemed necessary by the Editors, your manuscript will be sent back to one or more of the original reviewers for assessment. If the original reviewers are not available, we may invite new reviewers.

- Data accessibility

<http://datadryad.org/submit?journalID=RSOS&manu=RSOS-200473>

- Competing interests

- Authors' contributions

- Acknowledgements

- Funding statement

Kind regards,
Lianne Parkhouse
Editorial Coordinator

on behalf of Dr Seth Coffelt (Associate Editor) and R. Kerry Rowe (Subject Editor)
openscience@royalsociety.org

Associate Editor's comments (Dr Seth Coffelt):

Two reviewers agree that control theory was an interesting approach to gain insight into oncolytic viral therapy for cancer. One reviewer has some suggestions regarding the math, while another reviewer states that the conclusions are overstated. Please address these points.

Reviewers' Comments to Author:

Reviewer: 1

Comments to the Author(s)

The authors proposed an oncolytic virus therapy based on control theory. The model that was taken for the study was validated in a mouse model in previous published paper. Stability analysis was developed before. Unfortunately, I can not support the publication of the present study as it does not have a proper validation and/or comparison with any data.

Reviewer: 2

Comments to the Author(s)

see attachment

Reviewer: 3

Comments to the Author(s)

The application of proper quantitative analyses like control theory to therapeutic viruses is a good approach and advances the field.

However the abstract makes it sound like this paper has achieved far more than it really has, and could thus misinform. The authors muddle what is shown on one particular model with general statements about curing and controlling tumours; they mustn't. First, it is presented as if three treatments has been generally demonstrated to be a minimum, rather than one particular model requiring three to fit a specific criterion. If the authors want to claim "Our results demonstrate that at least three viral injections are required" they need a general technique that can fairly compare one- and two- treatment strategies, or they need to make it clear that the `_model_` is controllable. Similarly, if they want to claim "Our results demonstrate that at least three viral injections are required to control and reduce the tumor..." they need real experiments. The correct description would be that the models suggest or imply it.

Author's Response to Decision Letter for (RSOS-200473.R0)

See Appendix B.

RSOS-200473.R1 (Revision)

Review form: Reviewer 2

Is the manuscript scientifically sound in its present form?

Yes

Are the interpretations and conclusions justified by the results?

Yes

Is the language acceptable?

Yes

Do you have any ethical concerns with this paper?

No

Have you any concerns about statistical analyses in this paper?

No

Recommendation?

Accept as is

Comments to the Author(s)

This reviewer is satisfied with the revision.

Review form: Reviewer 3

Is the manuscript scientifically sound in its present form?

No

Are the interpretations and conclusions justified by the results?

No

Is the language acceptable?

No

Do you have any ethical concerns with this paper?

No

Have you any concerns about statistical analyses in this paper?

No

Recommendation?

Accept with minor revision (please list in comments)

Comments to the Author(s)

The authors have improved the text to clarify what has and hasn't been done.

For me, they haven't gone quite far enough - the statement in the abstract "When oncolytic virus therapy is performed using this feedback control of the tumor, the controller automatically tunes the dose of viral injections" still strongly implies to me that the authors HAVE performed the

virus therapy, rather than the counterfactual that future, possible virus therapy could be performed. This, too, needs moderating.

Similarly, in the following sentence, "feedback control delivers efficient and personalised dose of viral injections to achieve tumor regression better...", "delivers" needs to be replaced by "shows the potential to deliver".

Decision letter (RSOS-200473.R1)

05-Jun-2020

Dear Dr Rivadeneira:

On behalf of the Editors, I am pleased to inform you that your Manuscript RSOS-200473.R1 entitled "Oncolytic Virus Therapy Benefits from Control Theory" has been accepted for publication in Royal Society Open Science subject to minor revision in accordance with the referee suggestions. Please find the referees' comments at the end of this email.

The reviewers and Subject Editor have recommended publication, but also suggest some minor revisions to your manuscript. Therefore, I invite you to respond to the comments and revise your manuscript.

- Ethics statement

- Data accessibility

If you wish to submit your supporting data or code to Dryad (<http://datadryad.org/>), or modify your current submission to dryad, please use the following link:
<http://datadryad.org/submit?journalID=RSOS&manu=RSOS-200473.R1>

- Competing interests

- Authors' contributions

- Acknowledgements

- Funding statement

Because the schedule for publication is very tight, it is a condition of publication that you submit the revised version of your manuscript before 14-Jun-2020. Please note that the revision deadline will expire at 00.00am on this date. If you do not think you will be able to meet this date please let me know immediately.

on behalf of Dr Seth Coffelt (Associate Editor) and R. Kerry Rowe (Subject Editor)
openscience@royalsociety.org

Associate Editor Comments to Author (Dr Seth Coffelt):

Associate Editor: 1

Comments to the Author:

Dear Authors, please reword your statements as suggested by the Reviewer to make clear that you modeled oncolytic viral therapy instead of performed oncolytic viral therapy.

Reviewer comments to Author:

Reviewer: 2

Comments to the Author(s)

This reviewer is satisfied with the revision.

Reviewer: 3

Comments to the Author(s)

The authors have improved the text to clarify what has and hasn't been done.

For me, they haven't gone quite far enough - the statement in the abstract "When oncolytic virus therapy is performed using this feedback control of the tumor, the controller automatically tunes the dose of viral injections" still strongly implies to me that the authors HAVE performed the virus therapy, rather than the counterfactual that future, possible virus therapy could be performed. This, too, needs moderating.

Similarly, in the following sentence, "feedback control delivers efficient and personalised dose of viral injections to achieve tumor regression better...", "delivers" needs to be replaced by "shows the potential to deliver".

Author's Response to Decision Letter for (RSOS-200473.R1)

See Appendix C.

Decision letter (RSOS-200473.R2)

10-Jun-2020

Dear Miss Rivadeneira,

It is a pleasure to accept your manuscript entitled "Oncolytic Virus Therapy Benefits from Control Theory" in its current form for publication in Royal Society Open Science.

on behalf of Dr Seth Coffelt (Associate Editor) and R. Kerry Rowe (Subject Editor)
openscience@royalsociety.org

Appendix A

Referee report on 'Oncolytic virus therapy...'

The paper is in general nicely written. The main concerns are about some points in the Appendix which deserve some further precision.

- The controllability tests performed on page 18 are only sufficient since the model is nonlinear. The tests which are performed allow to conclude if the system is controllable thanks to the first order terms or not. If it is not, it may in general still be controllable due to higher order terms. Is it possible to check the accessibility property at equilibrium E_{q_h} (3)?
- last line on page 21: the difference between the set Y^{Tar} and the real value Cx is nonsense. Should one define a mean value of the target within the target set?
- last paragraph on page 21: in principle, a state feedback which achieves a pole placement, does not require any integral action. The steady state error can be set to zero with a proper state feedback. The integral action may eventually have some benefit w.r.t. robustness.
- ξ_{Tar} on page 21 is not defined in Table 5 (and not in paper either). k_i in equation (17) was denoted K_i in Table 5.

Some minor comments, mainly on typesetting:

- page 3, last paragraph: write 'the model fits well the time course' rather than 'the model fit well the time course'
- equation (1): write ' $r \log$ ' instead of ' $rlog$ '. By the way, in equation (6) and other places, the authors use the \ln notation.
- Above equation (9): write 'the Jacobian matrix is' instead of 'the Jacobians matrix is'.
- page 20, line 3: write 'Furthermore, we used' instead of 'Furthermore, We used'; line 4: write 'i.e. injections required to control' instead of 'i.e injections required control'
- Page numbers of ref [8] are 639-645. Page numbers of ref [18] are 233-241. Correct references of [21] are 2017 Feb;11(1):44-53.

Appendix B

Ms. Ref. No.: RSOS-200473

Title: Oncolytic Virus Therapy Benefits from Control Theory
Royal Society Open Science

Response to the Associate Editor

AE's comment. *Two reviewers agree that control theory was an interesting approach to gain insight into oncolytic viral therapy for cancer. One reviewer has some suggestions regarding the math, while another reviewer states that the conclusions are overstated. Please address these points.*

Authors' reply. *We thank the Associate Editor for supervising this review process. We are also grateful to the reviewers for their time and their constructive feedback. We have addressed the points made by the reviewers with the best of our knowledge. You will find our response to the comments of the reviewers listed in the next pages. We hope that this revised manuscript will meet the requirements for publication.*

Response to Reviewer 1

Reviewer's comment 1. *The authors proposed an oncolytic virus therapy based on control theory. The model that was taken for the study was validated in a mouse model in previous published paper. Stability analysis was developed before. Unfortunately, I can not support the publication of the present study as it does not have a proper validation and/or comparison with any data.*

Authors' reply. *We understand the concerns of the reviewer and we agree to tone down our claims as suggested by the other reviewers. We have revised the last paragraph of the discussion section for this matter. Yes, it is unfortunate not to be able to validate, compare and improve our work with experimental data at the moment. This is why we used a mathematical model which has been calibrated and validated using experimental data. This work is exploratory where our basic motivation is to highlight that control theory can add insight in this problem, and we hope this publication will foster productive collaborations with experimental groups working on oncolytic virus therapy to advance this field.*

Additionally, we have clarified in the abstract and introduction that the results presented in the paper are in-silico tested with the parameters identified from a previous experimental work in five nude mice.

In the discussion of the manuscript now reads:

In conclusion, this paper presents the first scheme to perform oncolytic virus therapy using impulsive control theory. Although we note caveats associated with experimental testing of the proposed protocols, these could be alleviated by including constraints related to viral doses and toxicity. Together, our findings advance knowledge on oncolytic virus therapy and open new avenues based on control theory to develop effective therapies. In future works, it is envisioned experimental trials to verify the analytical results developed here.

Response to Reviewer 2

The paper is in general nicely written. The main concerns are about some points in the Appendix which deserve some further precision.

Reviewer's comment 1. *The controllability tests performed on page 18 are only sufficient since the model is nonlinear. The tests which are performed allow to conclude if the system is controllable thanks to the first order terms or not. If it is not, it may in general still be controllable due to higher order terms. Is it possible to check the accessibility property at equilibrium Eqh(3)?*

Authors' reply. *The system is not accessible exactly at the healthy equilibrium because mathematically needs $S > 0$ to be accessible. But any small neighbourhood around $S = 0$ is accessible. In the manuscript, we have included the accessibility analysis of the model in the method section in the appendix. In addition, we presented the results of the accessibility and the corresponding analysis in the section 3 in the results.*

Accessibility characterization of oncolytic virus dynamics that was included in the manuscript:

“In this subsection, we illustrate the accessibility property of the model, but before this, the relative degree is computed to show that the output $h = \xi_1 + \xi_2$ is not an autonomous element for the system, and then at least 3 impulses are required to control the system.

So, applying the condition (ii) of Proposition 2 in [1], we get

$$\begin{aligned} \langle dh(\xi), g(\xi) \rangle &= \frac{\partial h(\xi)}{\partial \xi} g(\xi) = \begin{pmatrix} 1 & 1 & 0 \end{pmatrix} \begin{pmatrix} 0 & 0 & 1 \end{pmatrix}' = 0, \\ \langle dL_f h(\xi), g(\xi) \rangle &= L_g L_f h(\xi) = \begin{pmatrix} r(\ln\left(\frac{L}{\xi_1}\right) - 1) & -d_I & 0 \end{pmatrix} \begin{pmatrix} 0 & 0 & 1 \end{pmatrix}' = 0, \\ \langle dL_f^2 h(\xi), g(\xi) \rangle &= L_g L_f^2 h(\xi) = -\beta \frac{\xi_1}{\xi_1 + \xi_2} \left(d_I + r(\ln\left(\frac{L}{\xi_1}\right) - 1) \right). \end{aligned} \quad (1)$$

From equation (1), we can conclude that the impulse relative degree is $d^0(y) = 3$, as a matter of fact $\langle dL_f^2 h(\xi), g(\xi) \rangle \neq 0$ for any $\xi_1 > 0$, and $d_I \neq r(\ln\left(\frac{L}{\xi_1}\right) - 1)$, i.e the impulsive input affects the output $h(\xi) = \xi_1 + \xi_2$ after at least the third impulse, therefore $h(\xi)$ is not an autonomous element of the system. However, this adds a constraint in the reachable space, which becomes the strictly positive subspace of \mathbb{R}^3 . Now, From Theorem 2 in [1], we obtain

$$\begin{aligned} \text{ad}_f g &= \frac{\partial g}{\partial \xi} f - \frac{\partial f}{\partial \xi} g = \begin{pmatrix} \beta \frac{\xi_1}{\xi_1 + \xi_2} \\ -\beta \frac{\xi_1}{x_1 + \xi_2} \\ d_u \end{pmatrix}, \\ \text{ad}_f^2 g &= \frac{\partial \text{ad}_f g}{\partial \xi} f - \frac{\partial f}{\partial \xi} \text{ad}_f g = \begin{pmatrix} a_1(\xi_1, \xi_2) \\ a_2(\xi_1, \xi_2) \\ d_u^2 + \beta \alpha d_I \frac{\xi_1}{\xi_1 + \xi_2} \end{pmatrix}, \end{aligned}$$

where

$$a_1(\xi_1, \xi_2) = (\beta(\xi_2 r \ln\left(\frac{L}{\xi_1}\right) + \xi_1(d_u \xi_1 + (d_I + d_u)\xi_2 - (\xi_1 + \xi_2)r(\ln\left(\frac{L}{\xi_1}\right) - 1)))/(\xi_1 + \xi_2)^2,$$

$$a_2(\xi_1, \xi_2) = -((\beta(x_1(d_u(\xi_1 + \xi_2) + d_I(\xi_1 + 2\xi_2)) + x_2p(\xi_1)))/(\xi_1 + \xi_2)^2).$$

The determinant is

$$\det(g, \text{ad}_f g, \text{ad}_f^2 g) = -\beta^2 \frac{\xi_1^2}{(\xi_1 + \xi_2)^2} \left(d_I + r \left(\ln \left(\frac{L}{\xi_1} \right) - 1 \right) \right). \quad (2)$$

Eq. (2) shows that the dimension of $\overline{\dim \{g, \text{ad}_f g, \text{ad}_f^2 g\}}$ is equal to 3 for the same conditions of the relative degree. As a result, the strictly subspace \mathbb{R}_+^3 is accessible”.

Furthermore, we added the following sentences in the first paragraph of the discussion:

“The control analysis of the model suggests that the tumor is controllable after at least the third injections. This analytical result is consistent with the experimental findings in Kim J. H. et al, *Biomaterials*, 2011, because the experiments achieved tumor regression in some mice using three viral injections.”

[1] P. S. Rivadeneira and C. H. Moog, *Impulsive control of single-input nonlinear systems with application to HIV dynamics. Applied Mathematics and Computation*, 218:8462–8474, 2012.

Reviewer’s comment 2. last line on page 21: the difference between the set Y_{Tar} and the real value Cx is nonsense. Should one define a mean value of the target within the target set ?

Authors’ reply. The reviewer is right. There was a mistake in the definition of Y_{Tar} in Table 5 of the paper since Y_{Tar} is not a set, it is a target or set-point. This definition has been corrected in the table.

In the manuscript, we did not select a mean value within the target set $Z^{Tar} = \{T : 0 \leq T \leq 50\}$, instead we established the set-point $Z_{Tar} = 0$, which leads to $Y_{Tar} = Z_{Tar} - h(\xi_{ss}) = -h(\xi_{ss})$ for the linearized system.

The statement “ $Z_{Tar} = 0$ is chosen as a target or set-point within the therapeutic zone” has been added in the text to clarify that the objective used in the control strategy is a point and not a zone.

Reviewer’s comment 3. last paragraph on page 21: in principle, a state feedback which achieves a pole placement, does not require any integral action. The steady state error can be set to zero with a proper state feedback. The integral action may eventually have some benefit w.r.t. robustness.

Authors’ reply. The reviewer is right in his comment, but we did not explain properly the control problem for the linear system. The target for the nonlinear system is the healthy equilibrium, i.e. $Z_{Tar} = 0$. However, we used a linearized system around a different equilibrium $\xi_{ss} \neq Eq_h$ near from the healthy equilibrium to control the nonlinear system due to Eq_h is not controllable. Therefore, the target for the linear system is $Y_{Tar} = Z_{Tar} - h(\xi_{ss}) = -h(\xi_{ss}) = -Z_{ss}$. This implies that the control problem for the nonlinear system is a regulation problem, while for the linear system is a tracking problem. As a tracking problem, it is necessary either the integral action or the feed-forward gain.

We selected the integral action also to improve robustness of the therapy in presence of variations in the plant parameters, i.e. the biological rates.

For the sake of clarity, in the page 21, we have included the explanation regarding the regulation problem and the tracking problem for the nonlinear and linear system, respectively.

Reviewer's comment 4. ξ_{Tar} on page 21 is not defined in Table 5 (and not in paper either). k_i in equation (17) was denoted K_i in Table 5.

Authors' reply. ξ_{Tar} has been deleted because it is not necessary. The only concern for the control problem is the output, so we used Z_{Tar} instead.

k_i was corrected to K_i throughout all the manuscript.

Reviewer's comment 5. *Some minor comments, mainly on typesetting.*

Authors' reply. *All the minor comments were taken into account and corrected in the revised manuscript.*

Response to Reviewer 3

Reviewer's comment 1. *The application of proper quantitative analyses like control theory to therapeutic viruses is a good approach and advances the field.*

However, the abstract makes it sound like this paper has achieved far more than it really has, and could thus misinform. The authors muddle what is shown on one particular model with general statements about curing and controlling tumours; they must not. First, it is presented as if three treatments has been generally demonstrated to be a minimum, rather than one particular model requiring three to fit a specific criterion. If the authors want to claim "Our results demonstrate that at least three viral injections are required" they need a general technique that can fairly compare one- and two- treatment strategies, or they need to make it clear that the model is controllable. Similarly, if they want to claim "Our results demonstrate that at least three viral injections are required to control and reduce the tumor..." they need real experiments. The correct description would be that the models suggest or imply it.

Authors' reply. *We acknowledge the feedback of reviewer 3. We agree that our results apply to this model only. We have edited our abstract and the main body of the manuscript to revise overstatements.*

In the abstract we have clarified that the results are in-silico and for a particular model with parameters identified from a previous work in five nude mice. Additionally, we have replaced the words "our results demonstrate" by "the control analysis of the model suggests". The statement about in-silico results has been added in the introduction too.

In addition, at the end of the discussion section we stated that this paper presents the first scheme to perform oncolytic virus therapy using impulsive control theory, but the caveats associated with experimental tests, such as toxicity, should be further evaluated in the proposed protocols. We have also added that, for future works, we envision experimental tests to verify the analytical results developed here.

Appendix C

Ms. Ref. No.: RSOS-200473.R1

Title: Oncolytic Virus Therapy Benefits from Control Theory

Response to the Associate Editor

AE's comment: Dear Authors, please reword your statements as suggested by the Reviewer to make clear that you modeled oncolytic viral therapy instead of performed oncolytic viral therapy.

Authors' reply: We thank the Associate Editor and the reviewers for improving this manuscript with their suggestions. We have taken into account the new comments of the reviewers.

Response to Reviewer 2

Reviewer's comment: The authors have improved the text to clarify what has and hasn't been done.

For me, they haven't gone quite far enough - the statement in the abstract "When oncolytic virus therapy is performed using this feedback control of the tumor, the controller automatically tunes the dose of viral injections" still strongly implies to me that the authors HAVE performed the virus therapy, rather than the counterfactual that future, possible virus therapy could be performed. This, too, needs moderating.

Similarly, in the following sentence, "feedback control delivers efficient and personalised dose of viral injections to achieve tumor regression better...", "delivers" needs to be replaced by "shows the potential to deliver".

Authors' reply: As requested for the reviewer, we have modified the sentences in the abstract to clarify that we showed results from a simulation environment, and that we did not perform the virus therapy. Those sentences in the Abstract and the Author Summary was moderated.